# Determinants shaping the nanoscale architecture of the mouse rod outer segment

Matthias Pöge[1], Julia Mahamid[1†], Sanae S Imanishi[2], Jürgen M Plitzko[1], Krzysztof Palczewski[3]*, Wolfgang Baumeister[1]*

[1]Max Planck Institute of Biochemistry, Department of Molecular Structural Biology, Martinsried, Germany; [2]Eugene and Marilyn Glick Eye Institute and the Department of Ophthalmology, Indiana University School of Mediciney, Indianapolis, United States; [3]Gavin Herbert Eye Institute and the Department of Ophthalmology, Center for Translational Vision Research, Department of Physiology & Biophysics, Department of Chemistry, Department of Molecular Biology and Biochemistry, Irvine, United States

*For correspondence:
kpalczew@uci.edu (KP);
baumeist@biochem.mpg.de
(WB)

Present address: †Structural
and Computational Biology
Unit, European Molecular
Biology Laboratory, Heidelberg,
Germany

Competing interest: See page
22

Reviewing Editor: Giulia Zanetti,
Institute of Structural and
Molecular Biology, Birkbeck,
University of London, United
Kingdom

**Abstract** The unique membrane organization of the rod outer segment (ROS), the specialized sensory cilium of rod photoreceptor cells, provides the foundation for phototransduction, the initial step in vision. ROS architecture is characterized by a stack of identically shaped and tightly packed membrane disks loaded with the visual receptor rhodopsin. A wide range of genetic aberrations have been reported to compromise ROS ultrastructure, impairing photoreceptor viability and function. Yet, the structural basis giving rise to the remarkably precise arrangement of ROS membrane stacks and the molecular mechanisms underlying genetically inherited diseases remain elusive. Here, cryo-electron tomography (cryo-ET) performed on native ROS at molecular resolution provides insights into key structural determinants of ROS membrane architecture. Our data confirm the existence of two previously observed molecular connectors/spacers which likely contribute to the nanometer-scale precise stacking of the ROS disks. We further provide evidence that the extreme radius of curvature at the disk rims is enforced by a continuous supramolecular assembly composed of peripherin-2 (PRPH2) and rod outer segment membrane protein 1 (ROM1) oligomers. We suggest that together these molecular assemblies constitute the structural basis of the highly specialized ROS functional architecture. Our Cryo-ET data provide novel quantitative and structural information on the molecular architecture in ROS and substantiate previous results on proposed mechanisms underlying pathologies of certain PRPH2 mutations leading to blindness.

## Editor's evaluation

Pöge et al., present a study of the rod outer segment (ROS). These are specialised cilia of rod photoreceptor cells, essential for sensing light cues and initiating the vision process. The authors apply cryo-FIB milling to generate highly preserved rod samples and report high-quality cryo-tomographic data providing new insights into the ultrastructure of the ROS. The work reveals potential molecular scaffolds both in the lumen of the membrane stacks and on the surface of the stack providing the structural basis for ROS crucial ordered ultrastructure. The data presented here will be highly valuable for the field of phototransduction.

## Introduction

Rod photoreceptor cells are remarkably sensitive detectors for light. Phototransduction, the biochemical chain reaction that converts a light stimulus into a neuronal signal, is initiated when Rhodopsin (Rho) absorbs a single photon. Rho is a transmembrane G-protein-coupled receptor that resides in a specialized cellular compartment in rods, the rod outer segment (ROS), which contains a stack of hundreds of identically shaped and precisely spaced membrane disks. Each disk is composed of two parallel membranes connected at their periphery by a hairpin-like disk rim (*Sjostrand, 1953*). Dependent on the species, the disk membranes are only interrupted by one or more clefts, so-called disk incisures (*Makino et al., 2012*). Approximately 50% of the ROS disk membrane area is occupied by Rho (*Palczewski, 2006*). The light sensitivity of rods relies on the elaborate ROS membrane architecture, which seemingly evolved to maximize the surface area of the membranes, thereby maximizing the amount of Rho for photon capture (*Pugh and Lamb, 2000*).

The ultrastructure of ROS was described decades ago based on conventional transmission electron microscopy (TEM) (*Sjostrand, 1953*; ). The ROS disk stacks exhibit two key structural features: (i) the close packing of adjacent disks and (ii) the high curvature of the disk rims. The molecular determinants that give rise to this elaborate structural organization, however, remained largely elusive due to limitations in maintaining fine structural detail in preparations that require chemical fixation, dehydration, and heavy metal staining. Previous studies suggested that disk stacking may be assisted by the continuity of neighboring disk membranes (*Robertson, 1965*), while others suggested the existence of molecular connectors between adjacent disks (*Corless and Schneider, 1987*; *Roof and Heuser, 1982*; *Usukura and Yamada, 1981*), and between disk rims and the plasma membrane (PM) (*Goldberg et al., 2016*; *Nickell et al., 2007*; *Roof and Heuser, 1982*). Furthermore, EM studies of fixed and stained ROS identified a molecular assembly at disk rims referred to as the 'terminal loop complex' (*Corless et al., 1987*), which was proposed to enforce high membrane curvature. This complex is composed of a crescentic density located along the disk perimeter inside the disk lumen, which was observed to be linked by a transmembrane component to densities that connect neighboring disk rims. The 'terminal loop complex' appeared to form a 2D lattice. Yet, the molecular building blocks of the complex could not be identified with contemporary imaging technology. Mammalian ROS contain three abundant membrane proteins which localize exclusively to the disk rim: the rod cell-specific ATP binding cassette transporter ABCA4, which is important for the long-term viability of the retina (*Tsybovsky et al., 2013*), and the two small transmembrane proteins, peripherin-2 (PRPH2) (*Molday et al., 1987*) and ROS membrane protein 1 (ROM1) (*Bascom et al., 1992*). PRPH2 and ROM1 are homologs (*Kevany et al., 2013*) thought to associate non-covalently via dimers into homo- and hetero-tetramers (*Goldberg and Molday, 1996*), which form higher oligomers stabilized through disulfide bonds (*Loewen and Molday, 2000*). Isolated PRPH2-ROM1 complexes induce membrane curvature when reconstituted into lipid vesicles in vitro (*Kevany et al., 2013*), and heterologously expressed PRPH2 generates high curvature membranes in cells (*Milstein et al., 2017*). Models for membrane curvature formation have been proposed (*Milstein et al., 2020*) but no in situ structure of the ROS disk rim has become available.

Here, we sought to identify the key structural-molecular elements that support the formation of the mammalian ROS architecture, and to examine the validity of previous models for the native in situ structure. While many mutations leading to blindness are caused by distortions in ROS ultrastructure, or completely abolish ROS formation (*Boon et al., 2008*; *Daiger et al., 2013*), the underlying molecular mechanisms remain unresolved. Recent advances in cryo-ET (*Beck and Baumeister, 2016*; *Turk and Baumeister, 2020*) allowed us to obtain 3D molecular-resolution images of vitrified ROS in a close-to-native state providing further evidence for previously suggested mechanisms leading to ROS dysfunction.

## Results

### Defining the structural organization of ROS membranes

Mouse ROS were extracted using a fast retinal detachment method, minimizing structural deterioration. We employed a single mechanical disruption to detach ROS from the retina at the junction of their thin connecting cilium (CC). Examination of extracted ROS by light microscopy revealed intact ROS with the expected length on the order of 20 μm (*Nickell et al., 2007*; *Figure 1—figure*

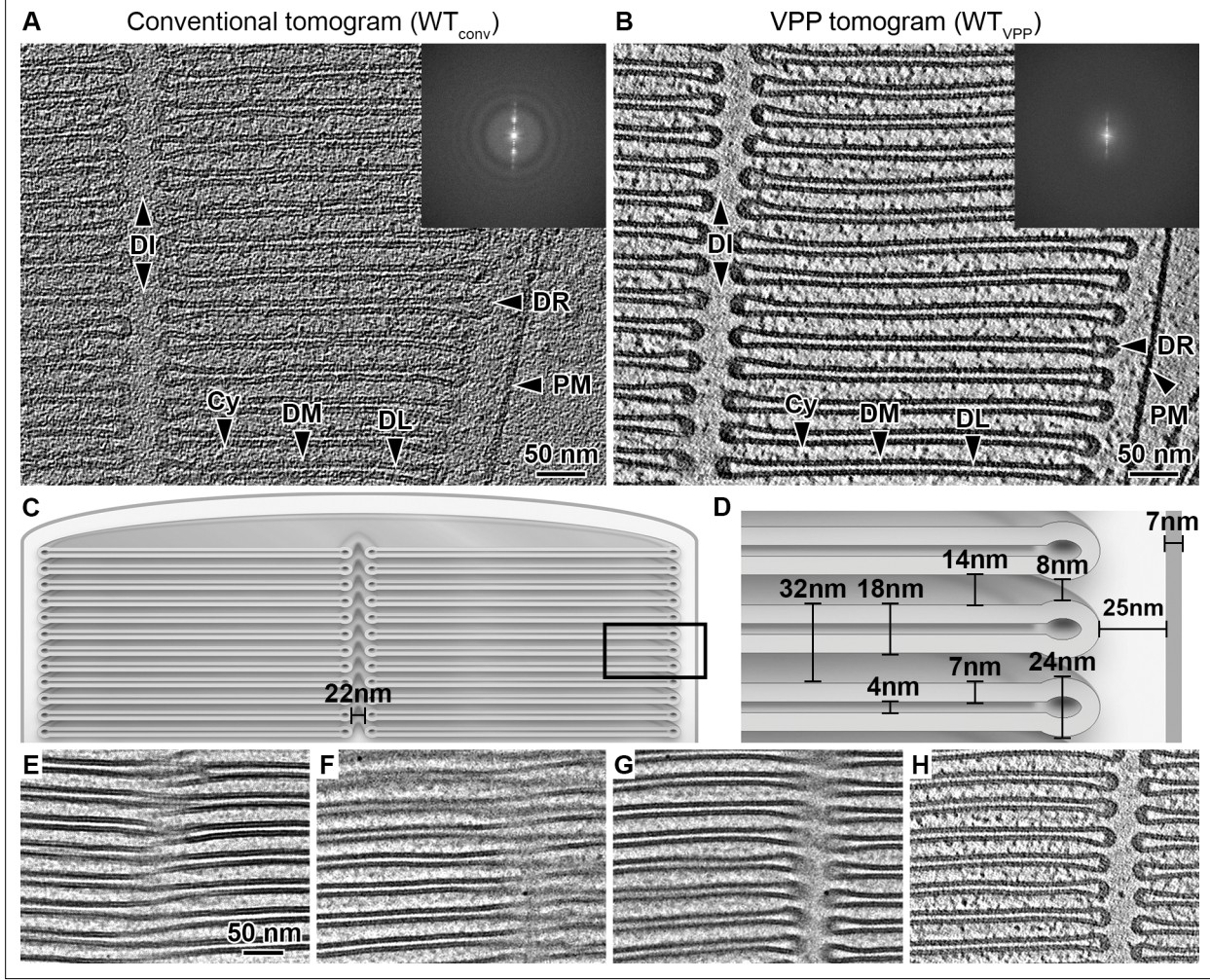

**Figure 1.** Quantitative characterization of ROS ultrastructure derived from cryo-ET. (**A**) Slice through a conventional tomogram acquired at 3 μm defocus and (**B**) in focus with Volta phase plate (VPP). Both imaging modalities allow distinction of the ROS membranes. The disk stack is composed of disk membranes (DM) surrounded by the disk rim (DR) and interrupted by the disk incisure (DI). The disk stack is enclosed by the plasma membrane (PM). DL denotes the disk lumen and Cy the cytosol. Insets: Fourier transforms of single projection images contributing to the tomograms. (**C–D**) Quantification of the characteristic ROS ultrastructure. The frame in (**C**) indicates the field of view in (**D**). (**E**) High-dose projection (~20 e⁻/Å²) showing a zipper-like structure. (**F**) Projection from a tomographic tilt-series (~1.4 e⁻/Å²) at tilt angle 25° showing a zipper-like structure similar to (**E**). (**G**) Projection at tilt angle 9°. (**H**) Tomographic slice reconstructed from the tilt-series. Zipper-like structure in (**F**) is resolved into the incisure.

The online version of this article includes the following figure supplement(s) for figure 1:

**Figure supplement 1.** Preparation of isolated mouse rod outer segments for cryo-ET.

**Figure supplement 2.** Transmission electron microscopy overview image of a lamella.

**Figure supplement 3.** Measurement of the repetitive distances between ROS disk membranes.

**Figure supplement 4.** Measurements related to the plasma membrane and the disk rim.

**Figure supplement 5.** Mean values and standard deviations (SD) of the measured ROS distances.

supplement 1A, B). The ROS suspensions were immediately applied to EM grids, vitrified by plunge freezing (*Figure 1—figure supplement 1C*, D) and thinned by cryo-focused ion beam (cryo-FIB) milling to create lamellae (*Schaffer et al., 2017*) with a thickness of around 150 nm (*Figure 1—figure supplement 1E*,F).

Lamellae typically contained several ROS as revealed by TEM overview images (*Figure 1—figure supplement 2A*). Conventional tomograms acquired with defocus reveal the highly ordered ROS membrane architecture (*Figure 1A*, *Video 1*). The use of the Volta phase plate (VPP) (*Danev et al., 2014*) further enhanced the contrast and allowed for the direct observation of cytosolic protein

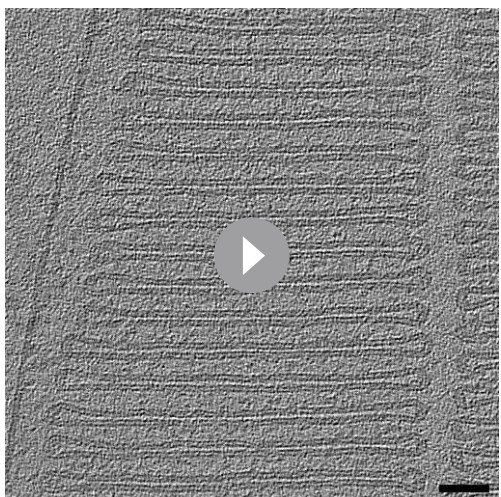

**Video 1.** Slices through a conventional tomographic volume acquired without Phase Plate and 3 µm defocus. Scale bar 50 nm.

https://elifesciences.org/articles/72817/figures#video1

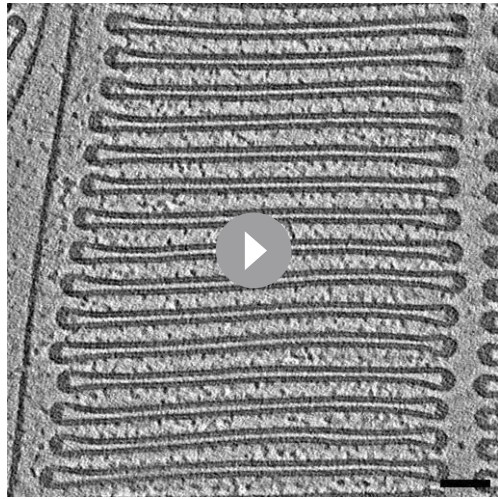

**Video 3.** Slices through the tomographic volume after weighted back projection of the tilt series in Video 2. In focus with Volta Phase Plate. Scale bar 50 nm.

https://elifesciences.org/articles/72817/figures#video3

densities (*Figure 1B*, *Videos 2 and 3*). Tomograms of ROS show that the supramolecular organization of the disk membranes exhibits the expected order over a range of micrometers. As this extended organization forms the foundation of ROS function in phototransduction, we derived a precise quantification of the structural parameters of the ROS ultrastructure (*Figure 1—figure supplements 3 and 4*). The resulting distances (*Figure 1C and D*, *Figure 1—figure supplement 5*) agree with previous studies of mammalian ROS (*Nickell et al., 2007*). The regular inter disk spacing is overall preserved across species (*Nickell et al., 2007*), and therefore defines with nanometer precision the 3D spatial framework within which fast molecular diffusion processes of phototransduction take place (*Calvert et al., 2001*).

It was previously suggested that the close proximity between disks is maintained by connectivity between the disk membranes, a model derived from 2D micrographs of metal-stained, plastic-embedded ROS sections showing zipper-like structures (*Robertson, 1965*). Cryo-EM 2D projections show similar patterns (*Figure 1E and F*). At a different tilt angle, however, the pattern is resolved into the disk incisure (*Figure 1G*). Tomographic reconstruction confirmed that the membranes at these positions are not interconnected (*Figure 1H*). We therefore conclude that the proposed model of continuous disk membranes is likely based on a misinterpretation of projection images that are disentangled by 3D imaging. Thus, we investigated next whether alternative structural elements contribute to the precise stacking of ROS disks.

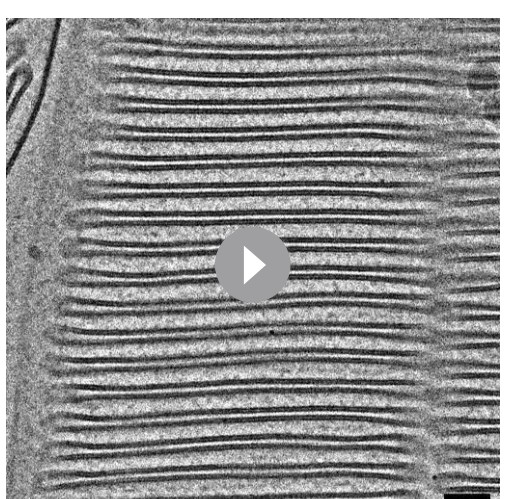

**Video 2.** Tilt series of a tomogram acquired in focus with Volta Phase Plate. Scale bar 50 nm.

https://elifesciences.org/articles/72817/figures#video2

## Segmentation reveals two distinct types of connectors between disks

An alternative mechanism proposed for disk stacking is the existence of molecular connectors between disks (*Corless and Schneider, 1987*; *Nickell et al., 2007*; *Roof and Heuser, 1982*; *Usukura and Yamada, 1981*). In agreement with these previous findings, we observed structures that connect membranes of two adjacent disks in the raw (*Figure 1B*) and filtered (*Figure 2A–C*) tomograms acquired with the VPP, but they were

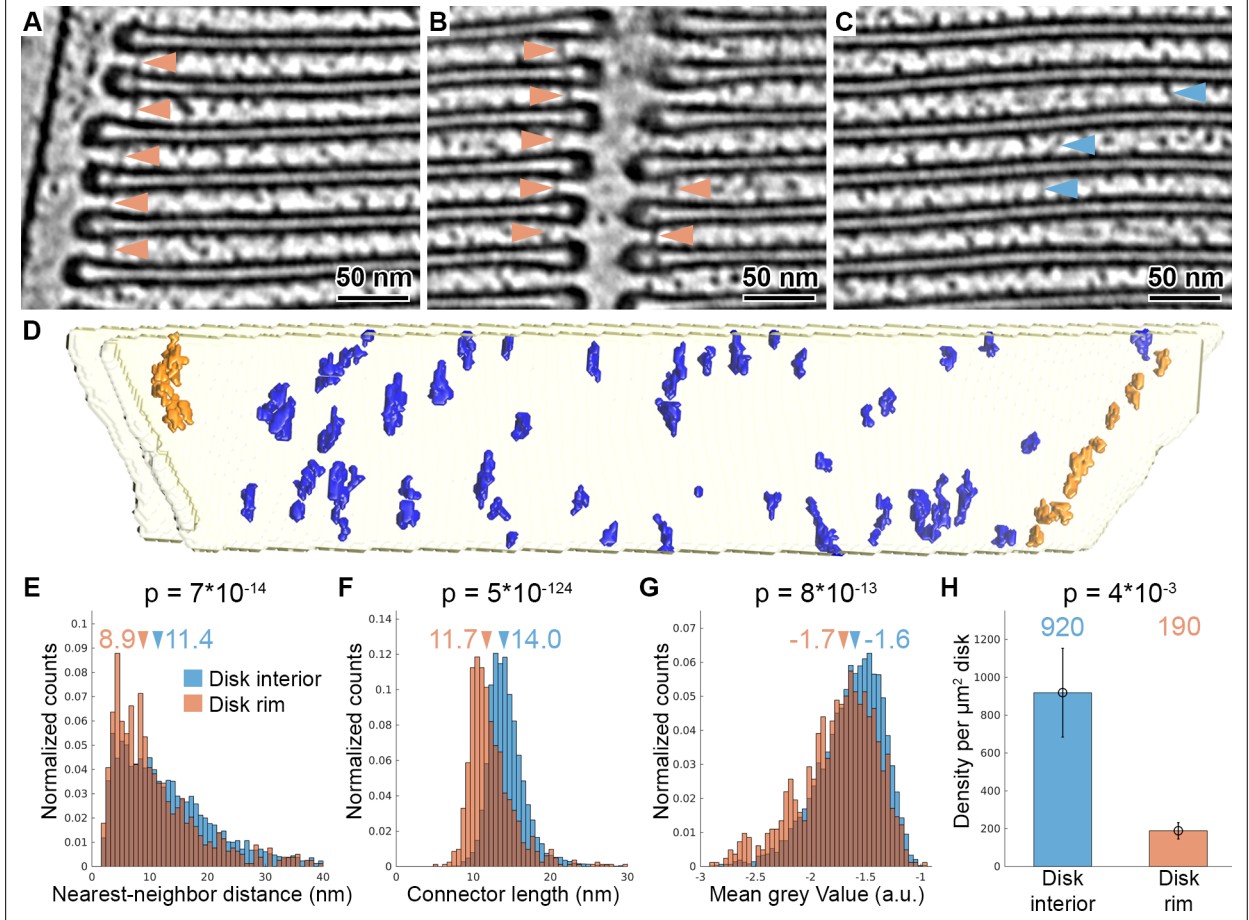

**Figure 2.** Tomography with VPP reveals molecular connectors between membranes of adjacent disks. (**A–C**) Slices of a tomogram acquired in focus with VPP (WT_VPP), filtered with a Gaussian Kernel (sigma = 4 voxel). Orange arrowheads in (**A**) and (**B**) indicate connectors localized at the disk rim in proximity to the plasma membrane and the disk incisure, respectively. Blue arrowheads in (**C**) point at connectors between the parallel membranes of adjacent disks in the disk interior. (**D**) Connectors segmented with the customized Pyto workflow on the example of one membrane pair viewed from the top (along ROS axis). Connectors within 40 nm of the outer disk periphery are defined as disk rim connectors (orange), and connectors in between the parallel membrane planes as disk interior connectors (blue). (**E–G**) Statistical analysis of 7000 connectors from five tomograms of the WT_VPP dataset. Histograms are shown of nearest neighbor distances (**E**), connector length (**F**) and mean gray value (**G**). Arrowheads above the histograms indicate the median values. (**H**) Mean value of connector density per μm² of total disk membrane determined in five tomograms (error bars: one standard deviation). p Values were calculated according to the two-sample Kolmogorov-Smirnov test.

The online version of this article includes the following figure supplement(s) for figure 2:

**Figure supplement 1.** Connector segmentation with the customized Pyto workflow.

**Figure supplement 2.** Comparison of segmentation methods for one membrane pair viewed from the top.

**Figure supplement 3.** Considerations for the statistical analysis and classification of connectors.

**Figure supplement 4.** Classification and averaging of disk connectors.

not discernible in the conventional tomograms (*Figure 1A*). Connectors were visualized in proximity to the disk rims, both at the outer periphery of the disks (*Figure 2A*, *Video 4*) and at the disk incisures (*Figure 2B*). In addition, densities that seem to connect the parallel membranes at the disk interior were found at lower frequency (*Figure 2C*, *Video 5*). Owing to the high contrast of the VPP data, the molecular connectors could be segmented in the raw tomograms.

We produced segmentations of the membrane bilayers (*Martinez-Sanchez et al., 2014*), based on which elements connecting two adjacent disks are defined, and used the Pyto software package (*Lučić et al., 2016*) to segment densities corresponding to these elements (*Figure 2—figure supplement 1A*). Here, we customized the original Pyto workflow by applying an additional mask prior to the segmentation to separate artificially continuous connectors (*Figure 2—figure supplement 1B*,

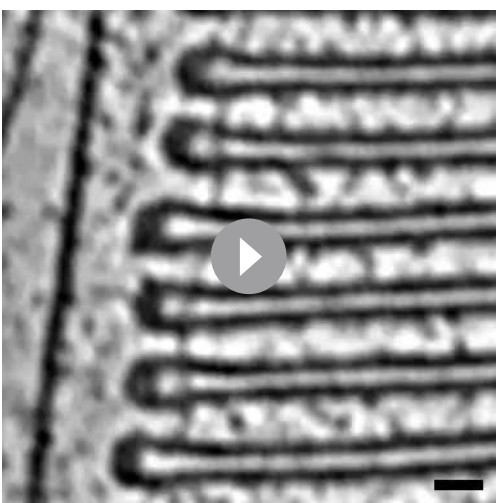

**Video 4.** Slices through a tomographic volume filtered with a Gaussian Kernel (sigma = 4 voxel). Shown are ROS disk rims in proximity to the plasma membrane. Many straight connectors between the disks can be observed at the disk periphery. Scale bar 20 nm.
https://elifesciences.org/articles/72817/figures#video4

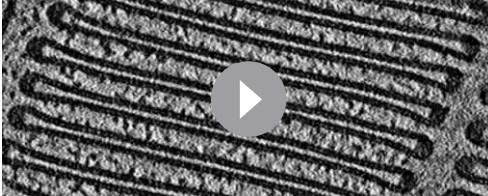

**Video 6.** Segmentation of connectors between two adjacent disks. Color code: yellow: membrane mask; blue: connectors in the disk interior; orange: connectors at the disk rim.
https://elifesciences.org/articles/72817/figures#video6

C) by water shedding (*Meyer, 1994*). The result of the automated segmentation method was compared to a manual segmentation (*Figure 2—figure supplement 2C*). Over 90% of the connectors were picked by both methods and the error of the determined coordinates was below 2 nm. Therefore, the automated segmentation allowed for quantitative analysis of connector abundance and arrangement in 3D.

Automated segmentation was performed on five tomograms resulting in the segmentation of 7000 connectors. Based on their proximity to the disk rim, they were divided into disk rim connectors, residing within 40 nm from the rim (*Figure 2—figure supplement 3A*), and disk interior connectors at the remaining membrane surface (*Figure 2D*, *Video 6*). Statistical analysis showed that rim connectors have shorter nearest-neighbor distances (*Figure 2E*), are shorter (*Figure 2F*), and have a lower grey value; that is, represent a more dense structure (*Figure 2G*) compared to the disk interior connectors. These results implied the existence of two distinct types of connectors. Based on the segmentation, we estimated the density of the connectors per surface area of disk to be 190 and 920 molecules per $\mu m^2$ of the disk membrane surface area for the rim and interior connectors, respectively (*Figure 2H*). By integrating available experimental data (*Batra-Safferling et al., 2006*; *Gilliam et al., 2012*; *Zhao et al., 2019*) and mass-spectrometry data (*Kwok et al., 2008*; *Skiba et al., 2013*), we tentatively assign disk rim connectors to glutamic-acid-rich protein 2 (GARP2), and the interior species to the enzyme phosphodiesterase 6 (PDE6). However, we were not able to validate this assignment experimentally. Due to the obvious structural flexibility of the connectors observed in the raw data (*Figure 2A–C*), we could not obtain further structural information by alignment, classification, and averaging of connector subvolumes (*Figure 2—figure supplement 4*).

## ROS disk rims are likely organized by a continuous scaffold of PRPH2-ROM1 oligomers

The second key structural feature of ROS architecture is the high curvature at the disk rims. There, the two membrane bilayers of a disk are connected *via* a hairpin-like structure, with a

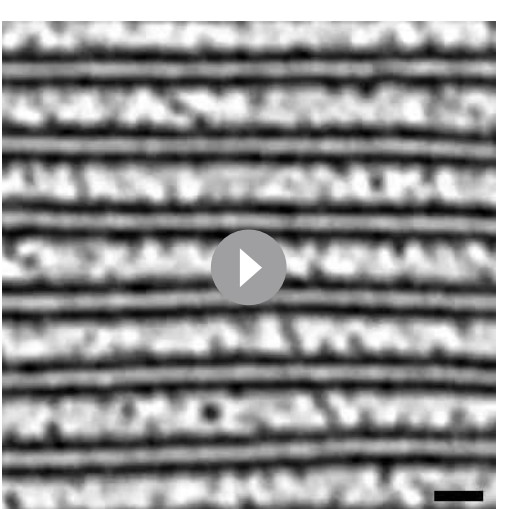

**Video 5.** Slices through a tomographic volume filtered with a Gaussian Kernel (sigma = 4 voxel). Shown are the parallel ROS disk membranes (disk interior). Structures that interconnect the disks can be found but are less abundant than at the disk periphery (Video 4). Scale bar 20 nm.
https://elifesciences.org/articles/72817/figures#video5

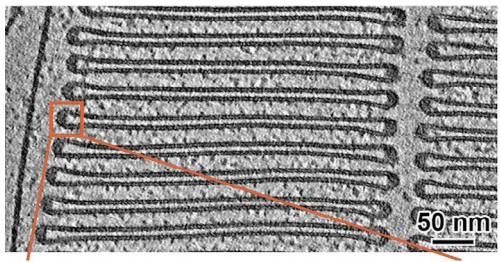

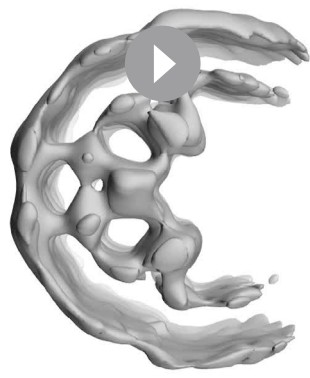

**Video 7.** Orientation of the disk rim subvolume average with respect to disk rims in the tomograms. The top panel shows a tomographic slice (WT$_{VPP}$) with one disk rim indicated by an orange frame. The bottom panel depicts the isosurface representation of the disk rim average obtained by focusing the alignment on the central density row. The view along the disk periphery reveals three luminal densities which form a continuous scaffold of three interconnected rows along the disk rim.

https://elifesciences.org/articles/72817/figures#video7

12 nm radius of curvature. To elucidate its organization in 3D, subvolumes were extracted at the disk periphery in the conventional dataset (WT$_{conv}$), aligned and averaged (*Figure 3—figure supplement 1*). The resulting average showed three densities (*Figure 3—figure supplement 1C*, *Video 7*) inside the disk lumen. In 3D, they form an ordered assembly of repeats organized in three interconnected rows, creating a continuous belt along the disk rim (focused on the central row in *Figure 3A*, *Video 8*). We measured a repeat length of 4.1 nm for all three rows in this average and a shift of half a repeat between the central and peripheral rows (*Figure 3—figure supplement 2C*). The view from the disk lumen onto the assembly appears to be C2 symmetric with respect to an axis perpendicular to the ROS axis.

The densities of the central row have a height of ~11 nm, stretching from the cytosol, through the membrane bilayer, and into the disk lumen (along the z-axis in *Figure 3B*). Slices taken at different heights reveal further structural features (*Figure 3C*). Each repeat contributes two small cytosolic densities. This 1.0 nm cytosolic domain is followed by a transmembrane domain extending over 6.0 nm with two diverging densities. These two densities then converge inside the disk lumen into a globular domain with a height of 2.6 nm. There, the densities within a repeat and its neighbors in the row come into close contact. Farther into the disk lumen, two diverging arms connect the central row to the peripheral rows. A repeat in the central row contacts two repeats, one on each of the peripheral rows located diagonally and inclined at an angle of ~63° with respect to the central repeat (*Figure 3—figure supplement 2B*). The head domain has a height of 1.5 nm

(*Figure 3B*). The shape and the dimensions (*Figure 3—figure supplement 3A*, B, *Video 9*) of repeats in the peripheral rows are similar to the central row; however, a pronounced density connects the peripheral repeats along the outside of the assembly (*Figure 3—figure supplement 3C*, *Video 10*).

Mammalian ROS contain three abundant proteins, which localize to the disk rim and harbor large disk luminal domains; namely, PRPH2 and ROM1 which form oligomers and ABCA4. To clarify the identity of the scaffold proteins, we analyzed the disk rims in VPP datasets of WT mice (WT$_{VPP}$) and of homozygote ABCA4 knockout mice (*Abca4$^{-/-}$*$_{VPP}$). The overall quality of the VPP averages is lower than for WT$_{conv}$, but cross-sections through the disk rim averages filtered to the same resolution reveal a similar structure in all three cases (*Figure 4*). Hence, the absence of ABCA4 has little impact on the architecture of the disk rim scaffold. Furthermore, the shape of the repeats is clearly different from the ABCA4 structure (*Liu et al., 2021*; *Tsybovsky and Palczewski, 2014*) while it agrees with the previously estimated dimensions of PRPH2-ROM1 tetramers (*Kevany et al., 2013*). However, a model of PRPH2 dimers predicted by AlphaFold2 (*Jumper et al., 2021*; *Mirdita et al., 2021*) matches the V-shape and the size of the repeats equally well (*Figure 3—figure supplement 5*, *Video 11*). Therefore, we hypothesize that the disk rim scaffold enforces the high membrane curvature at ROS disk rims and is composed of three interconnected rows of PRPH2-ROM1 oligomers. To test this hypothesis and to conclusively determine whether the repeats are PRPH2-ROM1 dimers or tetramers, a density map with near-atomic resolution would be needed.

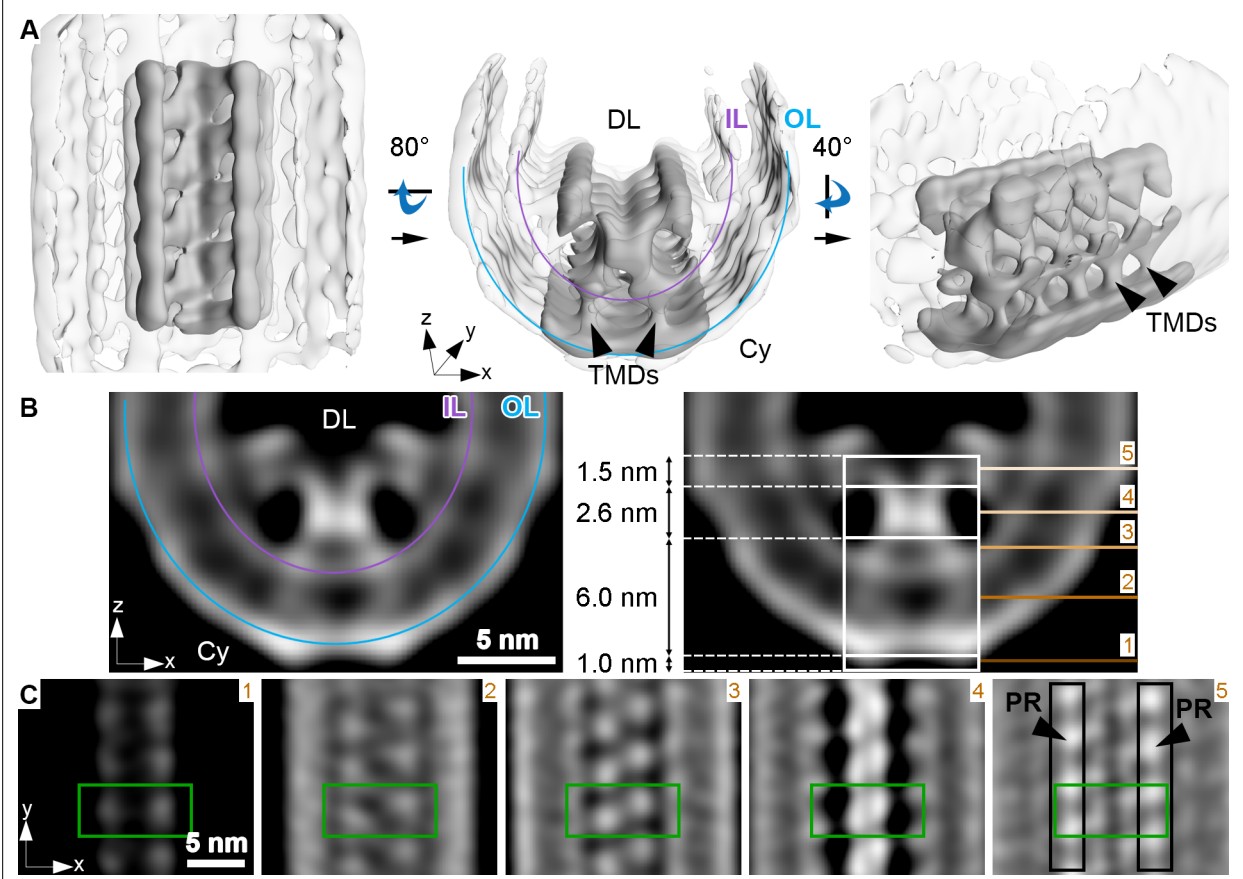

**Figure 3.** Membrane curvature at the disk rim is organized by a scaffold composed of three rows. This average was obtained by focusing the alignment on four repeats along the central density row (CD). (**A**) Isosurface representation of the disk rim subvolume average. The central row of density with its contacts to the peripheral rows is depicted in solid gray by applying the alignment mask to the average, and the signal of the whole disk rim average is shown in transparent grey. Black arrowheads indicate transmembrane densities (TMDs). DL denotes the disk lumen and Cy the cytosol. (**B**) Cross-sections through the disk rim average density without masking. (**C**) Orthogonal slices of the unmasked average at different z-heights. The green box is centered on the same repeat along the central density row throughout the slices. In the right panel, the signals of the peripheral rows (PR) are marked by black boxes. The locations of the slices are indicated by numbered lines in the right panel of (**B**). The signal of the inner leaflet (IL) and outer leaflet (OL) are indicated by a purple and a blue line, respectively.

The online version of this article includes the following figure supplement(s) for figure 3:

**Figure supplement 1.** Picking and alignment of disk rim subvolumes.

**Figure supplement 2.** Classification and analysis of the disk rim scaffold.

**Figure supplement 3.** Subvolume average of the peripheral rows in the disk rim.

**Figure supplement 4.** Fourier shell correlation and angular distribution of disk rim averages.

**Figure supplement 5.** Prediction of the PRPH2 dimer with ColabFold.

## Discussion

While the highly ordered ultrastructure of ROS was already described half a decade ago, its organization on the molecular level remained poorly understood. Here, we have utilized cryo-ET to obtain molecular resolution images of ROS and address open questions regarding the close disk stacking and the high membrane curvature at disk rims, which are specialized and essential structural characteristics of ROS. Based on these data, we provide an updated model for the structural organization of ROS (*Figure 5*). This advancement was enabled by the optimization of a ROS preparation method for cryo-ET that is gentler and faster than previously reported procedures (*Gilliam et al., 2012*; *Nickell et al., 2007*). Although some ROS were damaged during the preparation (*Figure 1—figure supplement 2A*), areas with structurally well-preserved ROS were easily identified and used exclusively for imaging.

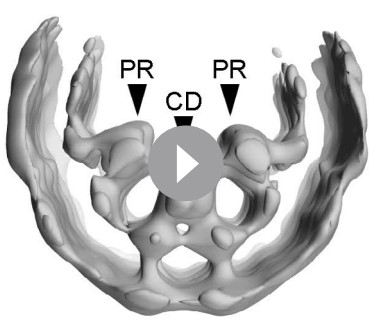

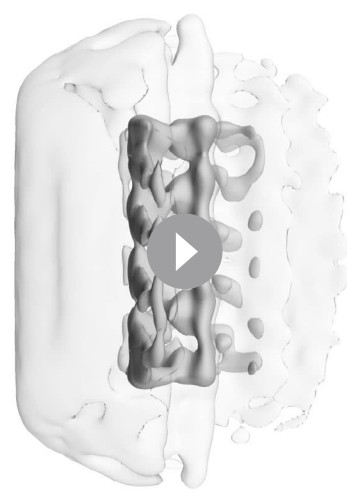

**Video 8.** Isosurface representation of the disk rim subvolume average. This average was obtained by focusing the alignment on four repeats along the central row of density. Initially, the whole, unmasked average is shown. Later, the central row of density (CD) with its contact to the peripheral rows (PR) is shown in solid grey by applying the alignment mask while the signal of the whole disk rim is displayed in transparent gray. The same representation was used in Figure 3A.
https://elifesciences.org/articles/72817/figures#video8

**Video 10.** Isosurface representation of the disk rim subvolume average for the peripheral rows (PR). This is the same average as in Video 9 with a similar representation, but at higher threshold emphasizing the density which links the repeats within the PR on the outside of the disk rim scaffold.
https://elifesciences.org/articles/72817/figures#video10

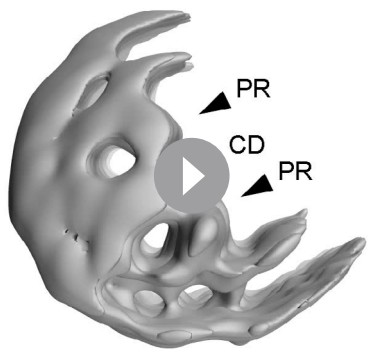

**Video 9.** Isosurface representation of the disk rim subvolume average for the peripheral rows (CW+ CCW). This average was obtained by centering the peripheral row (PR) in the subvolume box and focusing the alignment on four repeats along the PR. Initially, the whole, unmasked average is shown. Later, the peripheral row (PR) with its contact to the central density rows (CD) is shown in solid gray by applying the alignment mask while the signal of the whole disk rim is displayed in transparent gray. The same representation was used in Figure 3—figure supplement 3B.
https://elifesciences.org/articles/72817/figures#video9

Cryo-ET is a powerful tool to visualize the 3D molecular architectures of cells in a close-to-native state (*Figure 1*). Tomograms acquired with VPP exhibited enhanced contrast, revealing the molecular landscape in ROS and they enabled us to identify connectors between ROS disk membranes. Similar connectors have been observed previously (*Corless and Schneider, 1987*; *Nickell et al., 2007*; *Roof and Heuser, 1982*; *Usukura and Yamada, 1981*). The high quality of our data allowed us to quantitatively assess these connectors between disks, by segmenting them with a modified Pyto workflow, and their statistical analysis confirmed the existence of two distinct connector species. The disk rim connectors are shorter and more densely packed (*Usukura and Yamada, 1981*), while disk interior connectors are longer and fewer (*Kajimura et al., 2000*; *Figure 2*). Disk rim connectors were reported previously in frog ROS as part of the 'terminal loop complex' which appeared to be arranged in a 2D lattice (*Corless et al., 1987*). Our data suggest that the localization of disk rim connectors in mice is more variable because of their broad distribution of nearest-neighbor distances (*Figure 2E*) and the variable distances between disk rim connectors and the outer disk periphery (*Figure 2A and B*).

Based on previous studies combined with our quantitative analysis, we put forward a hypothesis for the molecular identity of the disk rim

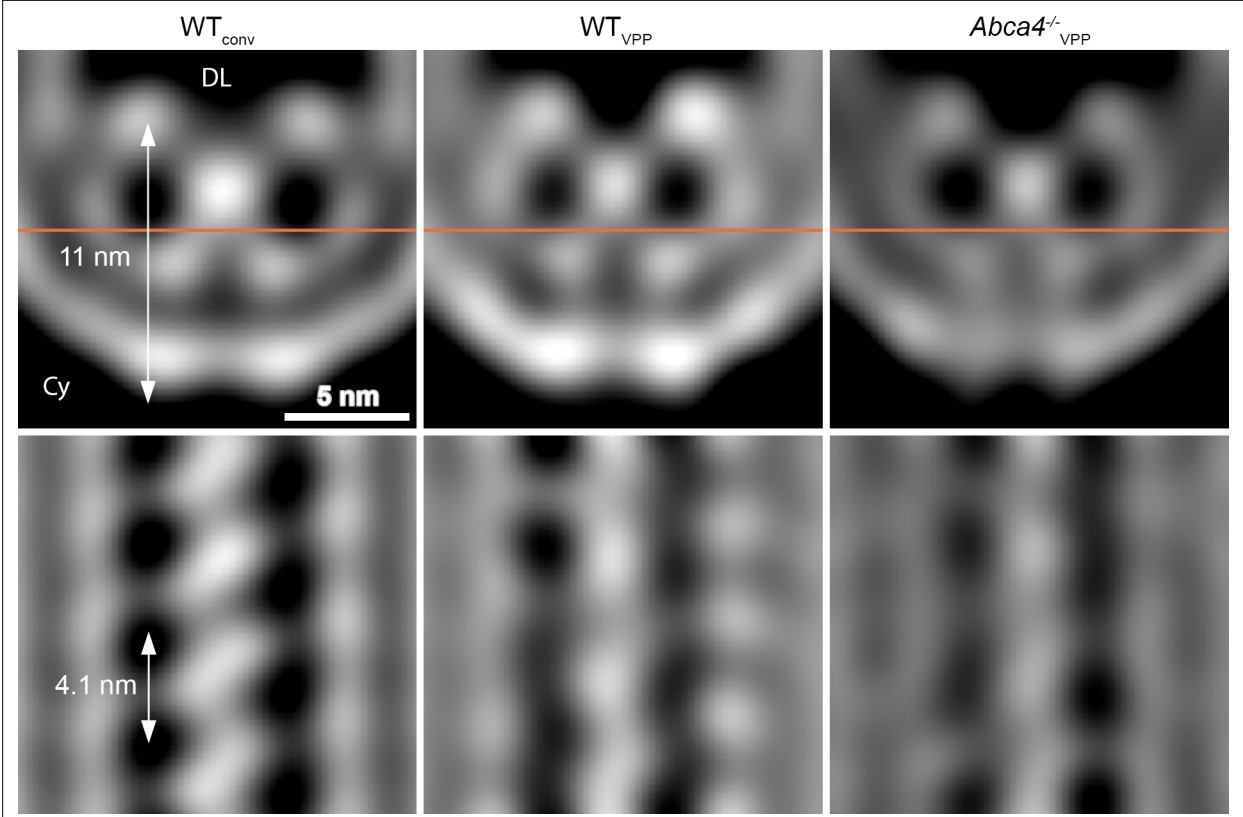

**Figure 4.** Comparison of the disk rim scaffold in WT and *abca4⁻/⁻* mice. Shown are orthogonal slices through the whole, unmasked averages of the central density (CD) in different datasets. All averages are filtered to a resolution of 30 Å. The left and right panel display the average of WT disk rims from a conventional (WT_conv) and a VPP dataset (WT_VPP). The right panel shows the VPP average derived from ABCA4 knockout mice (*Abca4⁻/⁻_vpp*). The orange line in the upper panels indicates the location of the slice in the bottom panels. DL denotes the disk lumen and Cy the cytosol.

connector which agrees in part with recent models. The *CNGB1* gene encodes three partially disordered glutamic acid-rich proteins (GARPs) (*Batra-Safferling et al., 2006*; *Colville and Molday, 1996*); namely, the β-subunit of the cyclic nucleotide gated cation channel (CNGC), and the alternatively-spliced proteins GARP1 and GARP2. All three GARPs are exclusively localized to disk rims (*Körschen et al., 1999*) or to the adjacent PM. CNGCs mediate the cation conductance of the ROS PM in response to light. GARP1 and GARP2 are soluble proteins which are tightly bound to membranes under physiological conditions by a hitherto unknown mechanism (*Körschen et al., 1999*). While loss of CNGCs has only a minor impact on ROS architecture (*Hüttl et al., 2005*), knockout of all three GARP isoforms destabilizes the diameter of the disks and results in the misalignment of disk rims (*Zhang et al., 2009*). This finding suggests a structural role of either GARP1, GARP2 or both. GARP1, however, is one order of magnitude less abundant in ROS than GARP2 (*Batra-Safferling et al., 2006*) and cannot account for the estimated concentration of 190 disk rim connectors per μm² disk. Thus, we suggest that GARP2 molecules form the connectors at the disk rim and organize the alignment of rims throughout the disk stack. It is likely that GARP2 molecules form an oligomeric state for the following reasons: (i) the hydrodynamic radius of GARP2 monomers is ~5 nm (*Batra-Safferling et al., 2006*) and therefore too small to establish ~12 nm long connectors measured at high precision in our data (*Figure 2F*); (ii) GARP2 is known to be in equilibrium with dimeric and tetrameric species in vitro (*Batra-Safferling et al., 2006*); and (iii) we observe less disk rim connectors than expected, considering that GARP2 occurs with approximately 500 molecules per μm² disk (*Batra-Safferling et al., 2006*; *Pugh and Lamb, 2000*).

Absence of all GARP isoforms does not abolish disk stacking (*Gilliam et al., 2012*), suggesting that the connectors in the disk interior can partially compensate for the loss of GARPs. The most plausible candidate for these connectors is the enzyme phosphodiesterase 6 (PDE6). The recently solved PDE6 structure (*Gulati et al., 2019*) shows that these 15 nm-long complex can bridge the

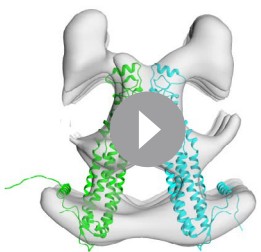

**Video 11.** Predicted model of a PRPH2 dimer docked into a repeat along the central row of density. The central row of density with its contact to the peripheral rows is shown in transparent grey by applying the alignment mask the whole disk rim average. The two PRPH2 chains within the dimer model are colored in green and cyan. The PRPH2-C150 cysteines are indicated as spheres in magenta.

https://elifesciences.org/articles/72817/figures#video11

14 nm gap between adjacent disks and its abundance matches the results of our segmentation (*Figure 2H*; *Pugh and Lamb, 2000*). To our knowledge, no other ROS protein exhibits the size and abundance required to contribute to the connectors visualized in our data (*Kwok et al., 2008*; *Skiba et al., 2013*). Further investigations will be required to probe this idea.

GARP2 was previously proposed to form connectors between rims of disks (*Batra-Safferling et al., 2006*) by directly interacting with PRPH2 (*Poetsch et al., 2001*). In contrast, we see no direct contact between the disk rim scaffold, which is likely composed of PRPH2-ROM1 oligomers, and disk rim connectors (*Figure 2—figure supplement 4A*, C). Furthermore, our subvolume average of disk rim connectors reveals no ordered transmembrane component linking the two structures (*Figure 2—figure supplement 4C*) as suggested previously (*Corless et al., 1987*). However, the quality of our averages is limited by the low number of a few hundred available subvolumes. The mechanism by which disk rim connectors bind to the membrane and what determines their contact site remain to be elucidated.

We could not observe connectors between disk rims and the plasma membrane in our tomograms as reported previously (*Nickell et al., 2007*; *Roof and Heuser, 1982*). These connectors were suggested to be formed by the GARP-part of the CNGC β-subunit (*Batra-Safferling et al., 2006*; *Poetsch et al., 2001*), interacting with PRPH2 in disk rims (*Pearring et al., 2021*; *Poetsch et al., 2001*). It is possible that the density of the intrinsically disordered GARP-part is not resolved as a clear connector in our tomograms. Furthermore, they would be expected to be present in only a few copies within the small field-of-view of our tomograms. Interestingly, while proteins for the experiments leading to the model were purified from mammalian sources (*Batra-Safferling et al., 2006*; *Poetsch et al., 2001*), the model assumes a distance of 10 nm between the PM and disk rims (*Batra-Safferling et al., 2006*). This value was derived from amphibians (*Roof and Heuser, 1982*) and deviates considerably from recent results 18 nm, (*Gilliam et al., 2012*) and from our current measurements in mice (~25 nm).

Our structural analysis of the ROS disk rim allowed us to resolve for the first time the crescentic rim density as part of the 'terminal loop complex' (*Corless et al., 1987*) into a continuous protein scaffold. This disk rim scaffold is composed of three interconnected rows with clear repeats inside the disk lumen (*Figure 3*). To our knowledge, only three proteins exist in ROS which are abundant, localized to the disk rim, and contain large disk luminal domains; namely, the homologs PRPH2 (*Molday et al., 1987*) and ROM1 (*Bascom et al., 1992*) which form noncovalent homo- and hetero-oligomers (*Goldberg and Molday, 1996*), and ABCA4 (*Papermaster et al., 1978*). Absence of ABCA4 does not significantly alter the structure of the disk rim scaffold (*Figure 4*), and we therefore suggest that given the compact and ordered structure of the scaffold, ABCA4 is likely not a component of it (*Figure 5C*). The luminal domain of ABCA4 is too large to be accommodated in the disk interior (*Liu et al., 2021*; *Tsybovsky et al., 2013*). Hence, we propose ABCA4 resides next to the scaffold at the disk rim. However, whether ABCA4 interacts with the rim scaffold or locates at a distance from it remains unclear.

Isolated PRPH2-ROM1 complexes and heterologously expressed PRPH2 induce membrane curvature in vitro (*Kevany et al., 2013*) and in cells (*Milstein et al., 2017*), respectively. Furthermore, the shape of the repeats appears to be similar to the dimensions reported for isolated PRPH2-ROM1

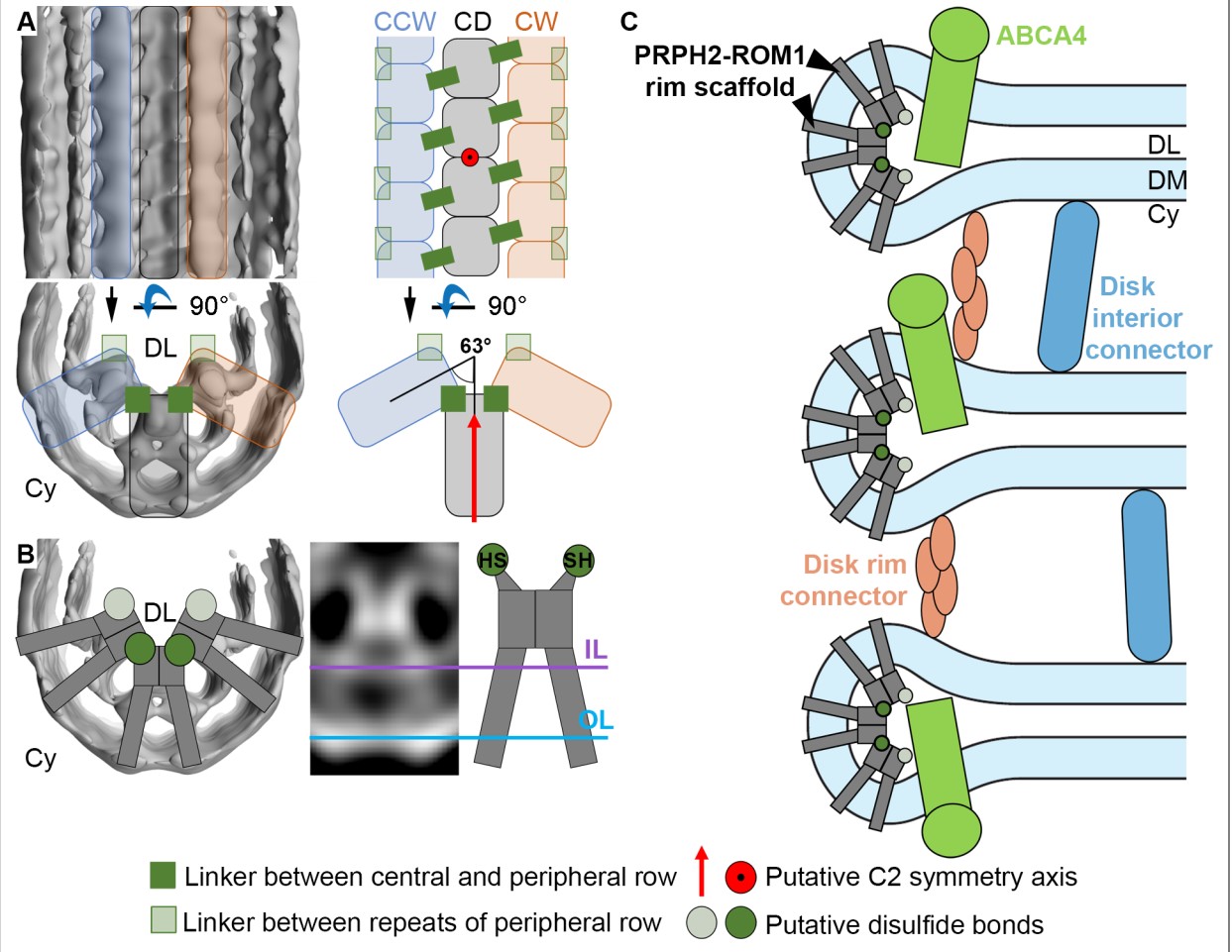

**Figure 5.** Models for the organization of ROS disk rims and the disk stack. (**A**) The general organization of the disk rim scaffold. (CD) marks the central density row, CW and CCW the clockwise- and counterclockwise peripheral row. (**B**) Non-covalently bound, V-shaped PRPH2-ROM1 complexes assemble into the disulfide bond-stabilized disk rim scaffold. We hypothesize that the PRPH2-C150 and ROM1-C153 cysteine residues which are responsible for intermolecular disulfide bonds are located in the head domain of the complexes forming the contacts between rows and repeats of the peripheral rows. (**C**) An updated model for the organization of the ROS disk stack. DL denotes the disk lumen, Cy the cytosol, and DM the disk membranes.

tetramers (*Kevany et al., 2013*). Recently, AlphaFold2 has proven its ability to predict the 3D structure of not only monomeric proteins with high accuracy (*Jumper et al., 2021*) but also that of small oligomeric complexes (*Evans et al., 2021*; *Mirdita et al., 2021*). An AlphaFold2 model predicted for the PRPH2 dimer closely resembles the V-shape and the size of the repeats observed in our disk rim average at a resolution of 18 Å (*Figure 3—figure supplement 5*, *Video 11*). The previously published low-resolution negative stain structure of PRPH2-ROM1 complexes interpreted as tetramers (*Kevany et al., 2013*) has approximately the same size as the predicted dimer structure. At this point, both observations are difficult to reconcile. Hence, we can only hypothesize that the repeats resolved in our average are smaller PRPH2-ROM1 complexes which oligomerize to form the disk rim scaffold and enforce the high membrane curvature at the disk rims. However, a density map with near-atomic resolution would be required to clarify whether the repeats are composed of PRPH2-ROM1 dimers or tetramers. At the current resolution, the disk rim scaffold appears to be C2 symmetric (*Figures 3A and 5A*), which is compatible with the predicted model of PRPH2 dimers (*Figure 3—figure supplement 5*) and the C2 symmetry axis found for the negative stain structure of putative PRPH2-ROM1 tetramers (*Kevany et al., 2013*).

Non-covalently bound PRPH2-ROM1 complexes are known to form higher order oligomers stabilized by disulfide bonds (*Loewen and Molday, 2000*), which are essential for normal disk morphogenesis (*Milstein et al., 2020*). The intermolecular disulfide brides are exclusively formed by the

PRPH2-C150 and ROM1-C153 cysteine residues which are located in the luminal domain (*Zulliger et al., 2018*). We hypothesize that these disulfide bonds (*Figure 5B*), are responsible for the contacts across rows (*Figure 3*) or between repeats of the peripheral rows (*Figure 3—figure supplement 3C*). Docking the predicted model of the PRPH2 dimer into the repeats of the central row, however, reveals that the two PRPH-C150 cysteine residues are not located where we observe these contacts but closer toward the inner membrane leaflet (*Figure 3—figure supplement 5B*, *Video 11*). This may be explained by errors in the prediction or the dimers assuming a different conformation upon oligo-merization and embedding into the highly curved membrane environment of the disk rim.

PRPH2-ROM1 oligomers isolated from native sources exhibit varying degrees of polymeriza-tion (*Loewen and Molday, 2000*) and ROM1 is excluded from larger PRPH2 oligomers (*Milstein et al., 2020*). We could not resolve this heterogeneity as additional structures to sufficient quality by subvolume averaging, but in combination with the inherent flexibility of the disk rim, it might be the reason for the restricted resolution of our averages. A model involving V-shaped PRPH2-ROM1 tetramers for membrane curvature formation was proposed recently (*Milstein et al., 2020*), but it comprises two rows of tetramers as basic building block of the disk rim scaffold which are linked in a head-to-head manner. Instead, our analysis in situ resolves three rows of repeats which are also linked by the luminal domain but are rather organized side-by-side (*Figure 5A*). Furthermore, our results raise the question whether the basic building blocks of the disk rim scaffold are PRPH2-ROM1 dimers or tetramers.

We propose a mechanism for disk rim curvature formation, where the two diverging transmem-brane densities of V-shaped PRPH2-ROM1 complexes displace lipids in the inner membrane leaflet. While unlinked complexes are able to induce some membrane curvature, their oligomerization into three continuous rows is required to force the membrane into this elongated, highly curved geometry (*Milstein et al., 2020*). Our data indicate that the luminal domains of complexes hold the disk rim scaffold together (*Figure 3C*), which is consistent with the fact that most pathological mutations of PRPH2 affect its luminal domain (*Boon et al., 2008*; *Goldberg et al., 2001*). In good agreement with previous work, it is possible that these mutations impair the formation of complexes and their disul-fide bond-stabilized oligomerization (*Chang et al., 2002*; *Conley et al., 2019*; *Zulliger et al., 2018*). Hence, these alterations could impede or completely prevent disk morphogenesis which, in turn, would disrupt the structural integrity of ROS, compromise the viability of the retina and ultimately lead to blindness.

## Materials and methods

### ROS extraction and cryo-preparation

Five- to 8-week-old wild type (WT) mice (C57BL/6 J, Jackson Laboratory, Bar Harbor, USA) and mice lacking the photoreceptor-specific ATP binding cassette transporter ABCA4 (*Abca4$^{-/-}$*) (*Weng et al., 1999*), were used for the isolation of rod outer segments (ROS). To minimize the interval between dissection and plunge-freezing, only one mouse was used for each preparation. The mouse was euth-anized by exposing it to $CO_2$ for 3–5 min followed by cervical dislocation. The first eyeball was excised with curved scissors and glued (Scotch Single-use super glue gel) with its sclera side down to a plastic Petri dish. The petri dish was filled with ice cold Ringer's buffer (10 mM Hepes, 130 mM NaCl, 3.6 mM KCl, 12 mM MgCl2, 1.2 mM CaCl2, 0.02 mM EDTA, pH 7.4) until the eyeball was fully covered. The eye was dissected as follows. First, a slit was made with a scalpel blade, and one blade of fine scis-sors inserted into the slit. The cornea was cut away and then the lens removed with fine forceps. To separate the retina from the retinal pigment epithelium, Ringer's buffer was applied gently between the layers with a P200 pipette. The retina was transferred into a 1.5 ml tube using a P1000 pipet. To prevent damaging the retina during transfer, the opening of the pipet tip was widened by cutting off its tip. The same procedure was applied to the second eye. After collecting two retinas in one tube, Ringer's buffer was removed and 25 µL of fresh Ringer's buffer added. The retinas were vortexed at 3200 rpm for 1 min to detach ROS. The sample was centrifuged at 100 rcf for 1 min at 4 °C using an Eppendorf 5415 R Centrifuge with an F 45-24-11 rotor. The centrifugation step enriched ROS in the supernatant which was transferred into a fresh tube (*Figure 1—figure supplement 1A*, B). To collect more ROS, 25 µL of Ringer's buffer were added to the retinas, which were then subjected to the same

ROS collection procedure. The combined supernatant was gently mixed by repetitive pipetting four times. The resulting sample was used for plunge-freezing. The total extraction time was 10–20 min.

For the light microscopy, 4 µL of the supernatant were placed on a clear bottom µ-dish (Ø = 35 mm, high, Ibidi GmbH, Gräfelfing, Germany). The images were taken on a CorrSight microscope (Thermo Fisher Scientific) operated at room temperature light path: wide-field, Objective: Zeiss EC Plan-Neofluar 40/0.9 NA Pol M27air objective (Zeiss, Oberkochen, Germany), working distance = 410 µm, Detector: Digital Camera C10600 ORCA-R2 (Hamamatsu Photonics Deutschland, Herrsching am Ammersee, Germany), image acquisition software: MAPS (version 2.1, Thermo Fisher Scientific).

For each glow-discharged copper grid (Quantifoil Cu 200 mesh, holy carbon film R2/1) 4 µl of the supernatant were applied. The grids were plunge-frozen in a liquid ethane/propane mixture (*Tivol et al., 2008*) at close to liquid nitrogen temperature using a Vitrobot Mark 4 (Thermo Fisher Scientific, Waltham, Massachusetts, USA). The blotting chamber conditions were set to 37 °C, 90% humidity, blot force 10 and 10 s blot time. The grids were blotted with a filter paper and a Teflon sheet from the reverse and front side, respectively. Grids were stored in liquid nitrogen until use.

Plunge-frozen grids were fixed into custom-made autogrids, mounted into a shuttle (*Rigort et al., 2010b*) and then transferred into a dual-beam focused ion beam and scanning-electron microscope (FIB/SEM, Quanta 3D FEG, Thermo Fisher Scientific) using a cryo-transfer system (PP3000T, Quorum Technologies, Lewes, UK). During FIB operation, samples were kept constantly close to liquid nitrogen temperature using an in-house-developed open nitrogen-circuit 360° rotatable cryo-stage (*Rigort et al., 2010a*). To improve sample conductivity and to reduce curtaining artifacts during FIB-milling, the samples were first sputter-coated with platinum in the Quorum prep-chamber (10 mA, 30 s) and then coated with organo-metallic platinum using an in situ gas injection system (GIS, Thermo Fisher Scientific) operated at 26 °C, at 12 mm stage working distance and 7 s gas injection time. Lamellae were prepared using a Gallium ion beam at 30 kV. FIB-milling was performed in a stepwise manner using rectangle patterns following similar procedures as in *Schaffer et al., 2017*. The initial step was conducted at a stage tilt angle of 25° with a beam current of 1 nA 10–20 µm away from the final lamella area. After rough milling, the stage was tilted to 20° and the ion current gradually reduced to lower currents as the thinning progressed (500 pA until 4 µm, 300 pA until 1 µm). For the final cleaning step, a low current of 50 pA was used to obtain lamellae thinner than 250 nm (*Figure 1—figure supplement 1D*, F). The progress of FIB-milling was monitored using the SEM operated at 10 kV and 42 pA (*Figure 1—figure supplement 1C*, E). For improved conductivity of the final lamella, the grid was again sputter-coated after cryo-FIB preparation with platinum in the Quorum prep-chamber (10 mA, 1 s) as previously reported in *Mahamid et al., 2016*.

## Cryo-transmission electron microscopy and tomography

Cryo-transmission electron microscopy observations were performed using a Titan Krios operated at 300 kV (Thermo Fisher Scientific). This microscope was equipped with a field-emission gun, a quantum post-column energy filter (Gatan, Pleasanton, USA), and a Volta phase plate (VPP, Thermo Fisher Scientific) (*Danev et al., 2014*). Bidirectional tilt-series were collected using SerialEM software (*Mastronarde, 2005*) between ±50° or ±60° starting at 20° with a tilt increment of 2° and a total exposure dose of ~100 e⁻/Å². The individual projection images were recorded as movies (dose fractionation mode) on a K2 Summit (Gatan) direct electron detector camera operated in counting mode with an image pixel size of 2.62 Å. The exposure dose for the projection at 0° $dose_{\alpha=0}$ was 1.6 e⁻/Å² fractionated over five frames. By acquiring more frames at higher tilt angles, the dose was adjusted as a function of the tilt angle α according to the following equation: $dose(\alpha) = dose_{\alpha=0} / \cos(\alpha)$.

A fraction of the tomographic tilt-series in this work were acquired with the VPP (*Danev et al., 2014*) and zero defocus (in focus). Alignment and operation of the Volta phase plate were carried out as described previously (*Fukuda et al., 2015*). During automated tilt-series acquisition an auto-focusing routine was performed using zero defocus offset with 5 mrad and 10 mrad beam tilt for conventional tilt series and data acquisition with VPP, respectively. For tilt series recorded in focus, the effect of the microscope's spherical aberration on the measured defocus was accounted for by setting the defocus target to 270 nm (*Danev and Baumeister, 2016*). Tomographic tilt-series were collected using standard automated acquisition procedures. All datasets are listed in *Table 1*.

Prior to the acquisition of the tilt-series, montage images at lower magnification (pixel size ~2 nm) were taken of the entire lamella. The montage tiles were aligned using the IMOD (version 4.10.18)

**Table 1.** List of used datasets.

| Dataset abbreviation | WT_conv | | WT_VPP | Abca4^{-/-}_VPP |
|---|---|---|---|---|
| Mouse sample | Wild type | | Wild type | *Abca4^{-/-}* |
| Volta phase plate | No | | Yes | Yes |
| Defocus (µm) | 3 | 4.5 | 0 | 0 |
| # Tomograms | 36 | 12 | 18 | 6 |
| EMPIAR accession code (EMPIAR-) | 10773 | | 10772 | 10771 |
| **Number of segmented connectors in five tomograms** | | | | |
| Disk rim connectors | - | | 800 | - |
| Disk interior connectors | - | | 6,200 | - |
| **Disk rim subvolumes for central density (CD)** | | | | |
| # all subvolumes | 53,000 | | 14,300 | 4,600 |
| # classified subvolumes | 9,000 | | 11,000 | 3,400 |
| Global resolution at FSC = 0.5 (Å) | 18.6 | | 22.5 | 27.5 |
| Global resolution at FSC = 0.143 (Å) | 16.9 | | 19.9 | 22.7 |
| Processing with Warp/*M* | Yes / Yes | | No / No | No / No |
| EMDB accession code (EMD-) | 13321 | | 13323 | 13324 |
| **Disk rim subvolumes for peripheral density (CW+ CCW)** | | | | |
| # all subvolumes | 106,000 | | - | - |
| # classified subvolumes | 48,000 | | - | - |
| Global resolution at FSC = 0.5 (Å) | 18.2 | | | |
| Global resolution at FSC = 0.143 (Å) | 16.8 | | - | - |
| Processing with Warp/*M* | Yes / Yes | | | |
| EMDB accession code (EMD-) | 13322 | | | |

(*Kremer et al., 1996*) command 'justblend'. Each lamella contained several ROS. In some cases, the ROS ultrastructure was partially distorted. As the distortions were locally confined, tilt-series were exclusively recorded in areas with ROS unperturbed by the sample preparation (*Figure 1—figure supplement 2A*). For each mouse strain and acquisition scheme, data was collected on samples derived from at least three different mice.

## Processing of tilt-series

Prior to tilt-series alignment, the projection images were corrected for beam-induced motion with MotionCor2 (*Zheng et al., 2017*). For the conventional dataset (no VPP and non-zero defocus), the CTF parameters of the projections were determined with Gctf (*Zhang, 2016*). Prior to tomogram reconstruction the projections were CTF-corrected with the IMOD function 'ctfphaseflip' and dose-filtered as described in *Grant and Grigorieff, 2015* with a MatLab implementation for tilt-series (*Wan et al., 2017*). Data acquired with VPP in focus was not CTF-corrected. Tilt-series alignment and tomographic reconstructions were performed using the IMOD (*Mastronarde and Held, 2017*) software package (version 4.10.18). Platinum particles originating from the protective platinum layer which were deposited over the lamella surface during FIB-milling served as fiducials (*Figure 1—figure supplement 2B*). Final alignment of the tilt-series images was performed using the linear interpolation option in IMOD. For tomographic reconstruction, the weighted back-projection algorithm in IMOD was used with the radial filter options left at their default values (cut off = 0.35; fall off = 0.05). In *Figures 2A–C and 4x* binned tomographic volumes (pixel size = 10.48 Å) were filtered by convolution with a Gaussian Kernel (sigma = 4 voxel) using the TOM toolbox (*Nickell et al., 2005*). Micrographs or

tomographic slices were visualized in IMOD (*Kremer et al., 1996*). In these cases, the density appears dark, that is with a low gray value.

## Distance calculation

To measure the repetitive distances of ROS disk membranes, contours of varying length perpendicular to the disk stack were defined in the disk interior (e.g. contour **h** of length h = ‖**h**‖ in *Figure 1—figure supplement 3A*). The contours were generated in 3dmod by opening the tomographic volume in the Zap window and creating a new model. Each contour included two points spanning across at least two ROS disks. Cuboids were cropped along these contours (*Figure 1—figure supplement 3B*). The base of the cuboids was square-shaped with an edge length a of 21 voxels (base edges labeled 'a' in *Figure 1—figure supplement 3B*). The cuboid voxels were averaged along the base area to obtain a 1D intensity profile of length h (*Figure 1—figure supplement 3C*). The distances were measured from the points where the membrane signals reached 50% of the maximum intensity (marked as red circles in *Figure 1—figure supplement 3C*).

For the thickness calculation of the plasma membrane (PM) $d_{PM}$, a total of 430 subvolumes were extracted from five tomograms along the PM, aligned, and subvolume averages were calculated for each tomogram. The PM thickness was determined in the 1D intensity profiles along **H** perpendicular to the PM where the signal was 50% of the maximum intensity (*Figure 1—figure supplement 4A*). A similar approach was used to compute the maximum diameter of the disk rim $d_{DR}$ parallel to the ROS cylinder axis. A total of 3000 subvolumes from six tomograms were aligned and averages calculated for each tomogram. The 1D intensity profile along **K** was used to determine $d_{DR}$ (*Figure 1—figure supplement 4B*). To calculate the width of the cytosolic gap at the disk incisure $d_{IN}$ and the distance between the PM and the disk rim $d_{PR}$, the refined coordinates of the disk rim subvolumes were utilized. The value of $d_{Shift}$ was determined as the distance from the center of the subvolume average to the outer periphery of the disk rim along **L** (*Figure 1—figure supplement 4B*). The subvolumes were separated into three groups: group 1 and group 2 comprised subvolumes on opposite sides of the disk incisure; group three contained subvolumes close to the PM. The gap at the incisure $d_{IN}$ was computed as follows: $d_{IN} = d_1 - 2 d_{Shift}$, where $d_1$ is the shortest distance of a coordinate in group one to a plane defined by its nearest neighbors in group two and vice versa (*Figure 1—figure supplement 4C*).

For the distance between PM and the disk rim $d_{PR}$, the central plane of the PM was segmented with TomoSegMemTV (*Martinez-Sanchez et al., 2014*). Then, $d_{PR}$ was calculated according to: $d_{PR} = d_2 - d_{Shift} - d_{PM}/2$, where $d_2$ is the shortest distance between a coordinate in group three and the central plane of the PM as determined by the segmentation (*Figure 1—figure supplement 4D*). Only the distance between neighboring disk rims (Distance G in *Figure 1—figure supplement 5*) was not directly measured in the tomograms but calculated as the difference between the unit cell distance and the maximum disk rim diameter (Distance B and H, respectively, in *Figure 1—figure supplement 5*). Besides subvolume averaging, the distance calculations and the required image processing steps were performed in MatLab aided by the TOM software toolbox (*Nickell et al., 2005*).

## Connector segmentation

The segmentations were performed on 4 x binned tomograms (pixel size = 10.48 Å). More dense structures, like proteins, appear darker in tomographic slices which translates into a lower gray value. First, all ROS membranes were automatically segmented by tensor voting (*Martinez-Sanchez et al., 2014*). The results of the automated segmentation and the original tomograms were loaded in Amira (v.6.2.0, Thermo Fisher Scientific). By comparing the two volumes, segmented patches which did not correspond to membranes were identified and manually removed. Afterwards, neighboring disk membranes of adjacent disks were grouped into pairs (*Figure 2—figure supplement 1C*). The results of the initial automated membrane segmentation correspond to the central membrane plane. By adding a layer of three voxels on either side of the central plane, the segmentation was grown to a thickness of 7 nm. This was then used to mask disk membranes with their apparent thickness of ~6.8 nm in raw tomograms. Additionally, these masks defined the borders of the cytosolic gap between disks which a connector must bridge. The cytosolic voxels between the membrane masks were normalized separately for each membrane pair to a mean value of zero and a standard deviation of one. This extinguished gradients in the gray value distribution throughout a tomogram caused

by heterogeneous lamella thickness and compensated for contrast differences between tomograms. To pick the connectors, the Pyto software package was used (*Lučić et al., 2016*). The original Pyto workflow segments connectors between the membranes of adjacent disks by evaluating all cytosolic voxels between the membrane masks as described below. The algorithm runs a gray value ramp from a user-defined lowest gray value $g_{min}$ to a highest gray value $g_{max}$, with a step size $g_{step}$ according to: $g_i = g_{min} + (i - 1)g_{step} \cap i = 1, 2, 3, …, g_{max}/g_{step}$.

At each iteration i, the algorithm performs a connectivity segmentation by selecting j groups of voxels $v_i^j$ based on four conditions:

1. all voxels in the group $v_i^j$ have a gray value smaller or equal to $g_i$,
2. the voxels of $v_i^j$ are in direct contact (face-to-face),
3. $v_i^j$ links the membrane masks of two adjacent disks,
4. no voxel of the group $v_i^j$ is in direct contact with any other voxel of a group $v_i^k$ with $k \neq j$.

During the next iteration with the gray value threshold at $g_{i+1}$, groups of voxels $v_{i+1}^j$ are selected that inevitably contain the $v_i^j$ with additional voxels of gray value $g_i < g(v_{i+1}^j - v_i^j) \leq g_{i+1}$ in direct contact with $v_i^j$. This defines a relationship among all connectors picked at the individual gray value steps. Connectors with $v_i^j \neq v_i^k$ are independent while connectors with $v_i^j \in v_{i+1}^j$ are related by an ancestor-descendant relation. The connector segmentation as output contains only independent groups of voxels which do not have ancestors. The original Pyto workflow is sketched in *Figure 2—figure supplement 1A*. Visual inspection of the segmented connectors and their comparison to the densities observed in the raw tomograms, however, revealed that fewer connectors with a higher volume than expected were segmented (*Figure 2—figure supplement 2A*, D). This difference is caused by several interconnected elements which were segmented as one connector.

Therefore, we customized the original Pyto workflow by applying an additional mask to the tomographic volume prior to the connector segmentation (*Figure 2—figure supplement 1B*). First, a binary mask was created that is one for all voxels with gray value below $g_{max}$, and elsewhere zero. Second, this binary mask was subjected to watershed transform (Fernand Meyer Algorithm *Meyer, 1994* implemented in MatLab) with 'catchment basins' filled from the center between the two membranes. Third, a volume with the watershed lines set to zero and elsewhere one was multiplied with the binary mask. The resulting mask was applied to the original tomographic volume. Then Pyto was used to segment connectors in the masked tomogram. A sketch of the customized Pyto workflow and its processing steps applied to the data for one membrane pair are depicted in *Figure 2—figure supplement 1B* and C, respectively. The threshold ramp for the original and the customized Pyto workflow was always started at the minimum gray value $g_{min}$ of –2 and ended at maximum gray value $g_{max}$ of –0.68 with a step size $g_{step}$ of 0.02.

The manual segmentation of connectors was performed as follows: initially, the membranes in the tomograms were masked as done for the automated segmentation. Tomographic volumes with the membrane mask applied were loaded into Amira (v.6.2.0, Thermo Fisher Scientific) and evaluated slice by slice. Groups of voxels that by visual inspection connect the membrane masks of adjacent disks were selected with the 'Magic Wand' tool (Amira v.6.2.0). The results of the connector segmentation and the membrane masks were visualized in UCSF Chimera (*Pettersen et al., 2004*).

To assess the quality of the customized Pyto segmentation approach, the results were compared to the manual segmentation (*Figure 2—figure supplement 2C*, D). Two major differences are apparent: First, the connectors selected automatically were bulkier than manually picked connectors. This is caused by the Pyto algorithm that picks voxels based on their gray value and their connectivity and evaluates all voxels at once, not in a slice-by-slice manner (*Figure 2—figure supplement 2E*). Second, fewer connectors were picked manually. This is likely due to inclined structures, which were not observed as continuous connectors in one single tomographic slice, but several successive slices. Consequently, they could be missed manually (*Figure 2—figure supplement 2E*). Therefore, picking of connectors with the automated segmentation approach is more reliable than the manual segmentation. Ninety percent of the connectors were picked by both methods and the error of the determined connector coordinates was below 2 nm. This error is small compared to the pixels size of 1 nm and the size of membrane patches with diameters of 500–1000 nm. Therefore, the shape of the automatically segmented connectors may not be reliable, but their abundance and arrangement in 3D can be quantitatively analyzed.

## Analysis of connector segmentation

A total of 7000 connectors were segmented in five VPP tomograms of wt ROS. The tomograms were selected based on a good IMOD tilt-series alignment scores and visual confirmation of well-resolved densities between ROS disks. The connectors and the membrane surface area were divided into two fractions. The disk rim fraction was within 40 nm from the outer periphery of disks rims. The remainder was considered the disk interior fraction. Based on this definition, 800 connectors were assigned as the disk rim connectors and 6200 as disk interior connectors. The local connector concentrations in the membrane fractions were calculated as the number of connectors $n_{fraction}$ per surface area $A_{fraction}$:

$$\rho_{fraction} = n_{fraction}/A_{fraction} \cap fraction = rim, interior.$$

To compare the determined local concentrations with literature values for ROS proteins, the connector concentrations per full disk membrane were calculated. The total disk membrane area $A^{tot}$ was estimated based on the morphological considerations specified in *Figure 2—figure supplement 3A* according to:

$$A^{tot} = \pi r_{out}^2 - r_{in}d_{cleft} = 1.3 \ \mu m^2.$$

The total area of the fractions per disk $A_{fraction}^{tot}$ were evaluated based on the distance threshold of 40 nm from the rim and the assumptions in *Figure 2—figure supplement 3A*:

$$A_{rim}^{tot} \approx \pi(r_{out}^2 - r_{in}^2) + 2d_{rim}r_{in} = 0.2 \ \mu m^2 \quad A_{interior}^{tot} = \pi r_{in}^2 - r_{in}(d_{cleft} + 2 \ _{drim}) = 1.1 \ \mu m^2.$$

The ratio $f_{fraction}$ of the total membrane area per fraction to the total disk area was calculated as: $f_{fraction} = A_{fraction}^{tot}/A^{tot} \cap fraction = rim, interior \ f_{rim} \approx 0.2 \ f_{interior} \approx 0.8$.

The connector concentration per disk is defined as:

$$\rho_{fraction}^{tot} = \rho_{fraction}f_{fraction}/2.$$

The division by two was introduced because a connector links two membranes. Therefore, the segmentation approach detects each connector effectively twice, in contrast to a density attached to only one membrane. The connector density was calculated for each tomogram separately.

To do the spatial analysis, each connector was assigned with a central coordinate $C_{con}$ located in the center between the two neighboring membranes (*Figure 2—figure supplement 3B*). A coordinate based on the center of mass of all connector voxels would result in off-center positions (*Figure 2—figure supplement 3B*) which would induce errors in the spatial analysis. Nearest-neighbor distances between connectors were calculated based on $C_{con}$. To estimate the connector length $L_{con}$, the two membrane contact points $P_{mb1}$ and $P_{mb2}$ of a connector with both disk membranes were determined (*Figure 2—figure supplement 3B*). $L_{con}$ was calculated as the sum of the distances between the central coordinate and the two contact points according to:

$L_{con} = ||\underline{C_{con}P_{mb1}}|| + ||\underline{C_{Con}P_{mb2}}||$, with $||\underline{C_{con}P_{mb1}}||$ and $||\underline{C_{Con}P_{mb2}}||$ denoting the distance between $C_{con}$ and the contact points $P_{mb1}$ and $P_{mb2}$, respectively (*Figure 2—figure supplement 3B*). The mean grey value was defined as the average gray value of all connector voxels. The statistical significance of differences between disk rim and disk interior connectors was established with the two-sample Kolmogorow-Smirnow test in MatLab.

## Subvolume averaging

Initially, binned subvolumes were extracted from dose-weighted and, if possible, CTF-corrected tomograms. The initial alignments were performed with scripts based on TOM, AV3 and Dynamo as described in *Schur et al., 2016*; *Wan et al., 2017*. The alignment references were exclusively derived from the data itself and low-pass filtered to 30 Å. To describe the orientation of subvolumes within the tomograms, triplets of Euler angles in 'ZXZ' convention were used, comprising the angles Phi, Theta, and Psi. Phi is the angle of the first in-plane rotation around the z-axis. Theta describes the second rotation around the new x-axis, and Psi the third rotation around the new z-axis. Classification of 3D subvolumes and the final alignments were performed in RELION (version 3) with 2 x binned or unbinned subvolumes. For the WT$_{conv}$ dataset, unbinned subvolumes were extracted with Warp (*Tegunov and Cramer, 2019*). Warp automatically generates a CTF model for each subvolume which is needed for RELION (*Tegunov and Cramer, 2019*). For VPP tomograms a simple 'fan'-shaped CTF model (*Bharat et al., 2015*) was created which was one for all information-containing slices in Fourier

space, and zero elsewhere. For the gray value representation of subvolume averages, the scale was inverted compared to the raw tomograms. Therefore, density in slices through subvolume averages appears bright, translating into a high grey value. Slices of subvolume averages were depicted in IMOD (*Kremer et al., 1996*), while isosurface representations and subvolume positions within the context of tomograms were displayed in UCSF Chimera (*Pettersen et al., 2004*).

## Subvolume averaging of connectors

Subvolume analysis of disk connectors was only performed in WT$_{VPP}$ tomograms. The initial subvolume extraction points of connectors were defined at their two membrane contact points P$_{mb1}$ and P$_{mb2}$ (*Figure 2—figure supplement 3B*) as elucidated by the segmentation. Initial Euler angles for Psi and Theta were determined so that the subvolume z-axis was parallel to the local normal vector of the disk membrane. The Phi angles were randomized. First, subvolumes were extracted from 4 x binned tomograms (pixel size = 10.48 Å, box size = 64$^3$) and aligned with shifts only allowed perpendicular to the membrane plane. For the disk interior connectors, the angle of the in-plane rotation was not searched, while for the disk rim connectors the whole 360° were covered to align the disk rims with respect to each other. The initial alignment brought the membranes into register and refined the initial orientations. The averages as a result of this alignment indicate a clear density protruding from the membrane into the cytosol, but it appears fuzzy compared to the membrane signal (*Figure 2—figure supplement 4A* and D, for disk rim and disk interior connectors, respectively). Probably, this is caused by the heterogeneity of the densities which could not be sorted by classification of 3D subvolumes in RELION. Hence, a different classification approach was chosen.

First, 4 x binned subvolumes were re-extracted at the refined positions (pixel size = 10.48 Å, box size = 32$^3$). Then, for each subvolume rotational averages around the z-axis were calculated and the resulting 2D images classified. For this, the 'plane_align_class.py' script was used as part of the PySeg package (*Martinez-Sanchez et al., 2020*). A cylindrical mask focused the classification on the cytosol between disks. Only a fraction of the subvolumes was assigned to classes with a clear connector-like density (40% and 6% for disk rim and disk interior connectors; *Figure 2—figure supplement 4B* and E, respectively), while many appear as false-positives (more than 30% and 60%; *Figure 2—figure supplement 4B* and D, respectively) because they have no or only a small membrane-attached density. Subvolumes of classes indicating a density between the membranes were considered most promising (*Figure 2—figure supplement 3C*). They were extracted from 2 x binned tomograms (pixel size = 5.24 Å, box size = 64$^3$) and aligned in RELION. The alignments were performed with the built-in sphere masks (diameter = 200 Å). The resulting averages, however, remained featureless and revealed no further structural insights (*Figure 2—figure supplement 4C*, F).

The high rate of putative false-positives indicated by the classification of the connector subvolumes suggests that our segmentation approach is error prone. Most likely, because the segmentation algorithm cannot distinguish two densities in close proximity protruding from opposite disks into the cytosol from an actual connector. On the other hand, we obtain classes with elongated densities that appear to link neighboring disks. Therefore, we assume that the two types of connectors indeed exist, yet at lower concentration than the initial segmentation determined (*Figure 2H*). Particularly, the disk rim connectors which are frequently seen in our tomograms (*Figure 2A and B*) and have been observed before (*Corless et al., 1987*; *Roof and Heuser, 1982*) are unlikely an artifact of the segmentation in the crowded environment of ROS.

## Subvolume averaging of disk rims

Subvolume analysis of disk rims was performed in WT$_{conv}$, WT$_{VPP}$ and a*bca4$^{-/-}$*$_{VPP}$ tomograms. Splines were manually picked along disk rims. For that, the tomographic volume was visualized in the 3dmod ZAP window, and a new model created. A new contour was defined for each disk rim by adding points along its outer periphery. Initial subvolume extraction points were set along the splines with 1 nm distance. Initial Euler angles for Psi and Theta were assigned so that the local spline direction dictated the orientation of the subvolume z-axis. The Phi angles of the in-plane rotation were randomized (*Figure 3—figure supplement 1A*). Initially, 4 x binned subvolumes (pixel size = 10.48 Å, box size = 64$^3$) were extracted. The initial average was composed of a strong density along the z-axis (*Figure 3—figure supplement 1A*). During the initial alignments, the translations along the spline were restricted to 1 nm and the entire Phi range was sampled, while the search range for Psi and Theta was restricted

to ±15°. Later, this search was refined. For the initial reference, a subset of 300 subvolumes was aligned against the unstructured, first average. After several iterations, the symmetry was broken until the average converged into the hairpin-like structure of the disk rim. This initial reference was then used to align the whole dataset. During this step, the subvolume positions converged to the disk rims and a first estimate for all three Euler angles was obtained (*Figure 3—figure supplement 1B*).

2 x binned subvolumes were extracted (pixel size = 5.24 Å, box size = $64^3$) at the refined coordinates and aligned. The average revealed a periodic scaffold with a repeat of ~4 nm (*Figure 3—figure supplement 1C*) and subvolume positions partially converging into the same points along the disk rim (lattice points) which had an average distance of ~4 nm (*Figure 3—figure supplement 1D*). This information was used to perform so-called distance-cleaning. At each of the lattice points, the particle with the highest similarity to the subvolume average, as estimated by the cross-correlation score, was kept and all others were discarded, which resulted in a minimal distance of 4 nm between subvolume coordinates. To potentially take the symmetry of the repeats into account, the subvolumes were reoriented by rotating the subvolume z-axis to point toward the disk center into the disk lumen, and the y-axis parallel to the ROS cylinder axis (*Figure 3—figure supplement 1C*).

The 4 x and 2 x binned subvolumes were aligned against references that were filtered to a resolution of 30 Å and all subvolumes were processed together. Only after distance cleaning, unbinned subvolumes (pixel size = 2.62 Å, box size = $128^3$) were extracted, split into half-sets, and independently processed in RELION. For this step, the tomograms of the WT$_{conv}$ dataset were preprocessed in Warp (version 1.0.9). Instead of using the entire preprocessing capabilities implemented in Warp, the motion corrected, non-CTF-corrected, and non-dose-filtered projections of the tilt-series were imported into Warp with the corresponding tilt-series alignment files. In the first step, the CTF parameters were calculated for each projection in Warp. The patch size for the CTF estimation was set to 512 × 512 pixels$^2$, the spatial frequency range used for the fit was between 34 Å and 12 Å and the defocus value was searched within ±2 μm of the tilt-series' target defocus. In a second step, the CTF was estimated for the whole tilt-series taking the tilted geometry of the individual projections into account. For this, the same settings as in the first CTF estimate were used, only the spatial frequency range was expanded to 7 Å.

As the previous alignment steps determined the orientations of the subvolumes with reasonable precision, the Euler angles in the RELION input star-file were set with a 'Prior' which allows restriction of the angular search around these angles. The alignment was focused on the central row of density with a wedge-shaped mask that covered four repeats (*Figure 3—figure supplement 1E*). After a first round of alignment, the subvolumes were classified without particle alignment, allowing 10–15 classes and 'regularization parameter T' was set to 0.1. Classes which indicated a highly ordered and symmetric disk rim scaffold were selected (*Figure 3—figure supplement 2A*), distance-cleaned, and separately aligned in RELION. The averages obtained by processing two independent half-sets of unbinned subvolumes were used to calculate Fourier shell correlation (FSC) curves (*Bharat and Scheres, 2016*). The global resolution was estimated as the spatial frequency where the FSC drops to 0.143 according to the 'gold-standard' (*Rosenthal and Henderson, 2003*), and according to the more conservative threshold of 0.5 (*Hrabe et al., 2012*). The final density maps were sharpened with a *B*-factor of –400 Å$^2$. The angular distributions of the final averages are depicted in *Figure 3—figure supplement 4C*. Throughout the processing, we did not distinguish between disk rims located at the outer disk periphery or at the disk incisure because at the current resolution the rim scaffold in these regions appears to be identical.

For WT$_{conv}$ further processing steps were applied. The output of the alignment with the classified subvolumes was imported into the *M* software (version 1.0.9 *Tegunov et al., 2020*). *M* performed a refinement of the tilt-series alignment. The default refinement parameters were used with an image and volume warp grid of 3 × 3 and 2 × 2 × 2 x10, respectively. Furthermore, particle positions and stage angles were refined but not the CTF estimate. Afterwards, all subvolumes before classification were re-extracted from tomograms with refined tilt-series alignment and processed by the same RELION pipeline as used before running *M*.

Even though the resulting CD average was obtained by focusing the alignment on four repeats along the central row of density, the average comprised the signal of the whole disk rim scaffold. We used different representations of this average to highlight certain aspects of the disk rim scaffold. The unmasked CD average shows the organization of the three interconnected rows of density (*Figure 3B*

and C and *Video 7*), while the alignment mask applied to the CD average reveals the signal of the transmembrane densities (*Figure 3A*, *Video 8*).

To elucidate the repeat length and the offset between the repeats of the peripheral and the central rows of density, the whole, unmasked CD average of the $WT_{conv}$ dataset was used. Three cuboid masks were defined around the luminal densities, one for each row (*Figure 3—figure supplement 2* left panel). The voxels inside the masks were then averaged perpendicular to the rows (*Figure 3—figure supplement 2C* right panel). The three resulting 1D intensity profiles were fitted to sine functions of the following shape:

$$y(x) = y_0 + A \sin(w_0 x + p).$$

The repeat length $\lambda$ is related to $w_0$ by:

$$\lambda = Ps\, 2\pi/w_0.$$

The offset between the rows $\Delta\lambda$ as:

$$\Delta\lambda = (p_{PR} - p_{CD})/2\pi.$$

Here, $p_{PR}$ and $p_{CD}$ denote the phase shift of the peripheral and central rows, respectively, and the pixel size Ps of the subvolume average was 0.262 nm. The determined repeat length 4.1 nm is further supported by the subvolume positions after alignment which converged into lattice points where the nearest neighbor distances had increased populations at multiples of ~4 nm (*Figure 3—figure supplement 1D*).

Furthermore, the peripheral rows of the $WT_{conv}$ dataset were analyzed. To generate the initial extraction points, the coordinates as result of the alignment of unbinned subvolumes for the central row were modified. First, the peripheral rows were centered and rotated to adopt a similar orientation as the central row before. Additionally, clockwise (CW) and counterclockwise (CCW) rows were aligned to each other. For the CW row, the orientation of CD was rotated by 63° around the y-axis. The CCW row required a 180° rotation of the CD orientation around the z-axis, followed by 63° around the y-axis (*Figure 3—figure supplement 2B*).

Subvolumes were extracted with Warp from unbinned tomograms and subjected to the same subvolume averaging pipeline as the central row. This included a first round of classification and alignment in RELION, tilt-series refinement in *M*, re-extraction of subvolumes from tomograms with refined tilt-series and a second round of classification and alignments in RELION. The alignment mask focused on four repeats along the peripheral row. All alignment steps were performed for CW and CCW separately, and both peripheral rows combined (CW + CCW). The global resolution of CW + CCW was with 18.2 Å slightly higher in comparison to the individual rows (*Figure 3—figure supplement 4A*). The symmetry operation required to combine the CW and CCW as well as the higher quality of the combined average further indicate the C2 symmetry of the disk rim scaffold. The putative C2 symmetry, however, was not applied during subvolume analysis of the central row. The averages in *Figure 4* were filtered to a resolution of 30 Å using the TOM function tom_filter2resolution (*Nickell et al., 2005*).

Our best disk rim averages have a rather low resolution (~18 Å) given the number of used subvolumes (*Table 1*) compared to previous results for other protein scaffolds (*Dodonova et al., 2017*; *Schur et al., 2016*). This is probably caused by the flexibility and the heterogeneity which is characteristic of many structures in cells. The flexibility of the disk rim scaffold is indicated by the varying rim diameters measured in the tomograms (*Figure 1—figure supplement 5*). We tried to avoid the influence of flexibility by focusing the alignment on individual repeats instead of several repeats along a row. This did not improve the global resolution suggesting that the repeat itself is heterogeneous or flexible.

## Structural prediction of PRPH2 oligomers

To further improve our model of the disk rim scaffold, we modeled 3D structures of PRPH2. Since there are no available homolog structures in PDB for conventional homology-based structure prediction, we relied on ColabFold (*Mirdita et al., 2021*), a notebook environment based on AlphaFold2 (*Jumper et al., 2021*). For the prediction, we used the sequence of mouse PRPH2 available in the UniProt database (P15499) and the default settings in the following, publicly available notebook: https://colab.research.google.com/github/sokrypton/ColabFold/blob/main/beta/AlphaFold2_advanced.ipynb.

The resources provided by ColabFold allowed us to calculate predictions for PRPH2 monomers and dimers (*Figure 3—figure supplement 5A*) but not for tetramers due to memory requirements. The three domains of PRPH2 can be readily assigned: The partially disordered, cytosolic C-terminus (*Milstein et al., 2017*; *Ritter et al., 2005*) exhibits the lowest predicted local distance difference test (pLDDT) score across the sequence. It is followed by the transmembrane domain composed of four transmembrane segments typical for proteins of the tetraspanin family (*Termini and Gillette, 2017*). The third domain resides inside the disk lumen and is dominated by the large extracellular loop 2 (EC2), which is also characteristic for tetraspanins (*Termini and Gillette, 2017*). Of the five predicted PRPH2 dimer models, four models were V-shaped and resembled the shape of the repeats resolved in our disk rim average. The best scoring model was docked into one repeat of the central density row of our disk average by ridged-body fitting using the fitmap command in Chimera (*Figure 3—figure supplement 5B*, *Video 11*). Apart from the disordered C-terminal regions, this dimer model fits well into the repeats of the disk rim scaffold and appears to be C2-symmetric which is compatible with the putative C2 symmetry that we suggest for the disk rim scaffold. A similar implementation of Alpha-Fold2 on the computer cluster at the EMBL in Heidelberg could circumvent the memory issues and predict models for PRPH2 tetramers. None of the five predicted models, however, fit into either one single or two neighboring repeats of the disk rim scaffold because the tetramers were too large and of different shape. This result further supports the idea that the repeats are indeed PRPH2-ROM1 dimers and not tetramers.

## Acknowledgements

We are grateful to Z Dong for providing the *Abca4*[-/-] mice, to A Martinez-Sanchez and V Lučić for assisting with the segmentation tools, to W Wan and S Khavnekar for streamlining tilt-series preprocessing, to W Wan and F Beck for advice on subvolume averaging, and to P Erdmann, G Pfeifer and M Schaffer for support on the electron microscopes. This work was supported by grants from the CIFAR program Molecular Architecture of Life for WB and KP, by funding from the National Institutes of Health: R01EY030873 (KP), R01EY030912 (KP), and unrestricted grants from Research to Prevent Blindness to the Department of Ophthalmology at UCI. MP acknowledges the support of the Max-Planck Institute of Biochemistry and JM the support from EMBL.

## Additional information

### Competing interests

Julia Mahamid, Sanae S Imanishi, Krzysztof Palczewski: is Chief Scientific Officer of Polgenix Inc. Wolfgang Baumeister: holds a position on the advisory board of Thermo Fisher Scientific. The other author declares that no competing interests exist.

### Funding

| Funder | Grant reference number | Author |
|---|---|---|
| National Institutes of Health | R01EY030873 | Krzysztof Palczewski |
| National Institutes of Health | R01EY030912 | Krzysztof Palczewski |
| Max Planck Institute for Dynamics of Complex Technical Systems Magdeburg | Open-access funding | Matthias Pöge |

The funders had no role in study design, data collection and interpretation, or the decision to submit the work for publication.

### Author contributions

Matthias Pöge, Conceptualization, Data curation, Formal analysis, Investigation, Methodology, Software, Validation, Visualization, Writing – original draft, Writing – review and editing; Julia Mahamid,

Conceptualization, Data curation, Formal analysis, Investigation, Methodology, Project administration, Supervision, Validation, Writing – original draft, Writing – review and editing; Sanae S Imanishi, Conceptualization, Data curation, Methodology, Writing – review and editing; Jürgen M Plitzko, Krzysztof Palczewski, Wolfgang Baumeister, Conceptualization, Methodology, Project administration, Resources, Supervision, Writing – review and editing

### Author ORCIDs
Matthias Pöge (iD) http://orcid.org/0000-0003-2331-4638
Julia Mahamid (iD) http://orcid.org/0000-0001-6968-041X
Sanae S Imanishi (iD) http://orcid.org/0000-0001-9884-2123
Jürgen M Plitzko (iD) http://orcid.org/0000-0002-6402-8315
Krzysztof Palczewski (iD) http://orcid.org/0000-0002-0788-545X
Wolfgang Baumeister (iD) http://orcid.org/0000-0001-8154-8809

### Ethics
The animals used were bred for scientific purposes. At the University of California, Irvine, all mice were housed in the vivarium where they were maintained on a normal mouse chow diet and a 12 h / 12 h light / dark cycle. All animal procedures were approved by the Institutional Animal Care and Use Committee (IACUC, AUP-21-096) of the University of California, Irvine, and were conducted in accordance with the Association for Research in Vision and Ophthalmology Statement for the Use of Animals in Ophthalmic and Visual Research. The research at the Max Planck Institute of Biochemistry did not involve experiments on an animal. All animals were sacrificed prior to the removal of organs in accordance with the European Commission Recommendations for the euthanasia of experimental animals (Part 1 and Part 2). Breeding and housing as well as the euthanasia of the animals are fully compliant with all German (e.g. German Animal Welfare Act) and EC (e.g. Directive 86/609/EEC) applicable laws and regulations concerning care and use of laboratory animals. The Max Planck Institute of Biochemistry has a licence for breeding and housing of laboratory animals which includes the killing of animals solely for the use of their organs or tissues (No.4.3.2-5682/MPI für Biochemie - rural districts office).

### Decision letter and Author response
Decision letter https://doi.org/10.7554/eLife.72817.sa1
Author response https://doi.org/10.7554/eLife.72817.sa2

---

## Additional files

### Supplementary files
• Transparent reporting form

### Data availability
The subvolume averages of ROS disk rims are deposited in EMDB under the accession codes: EMD-13321, EMD-13322, EMD-13323 and EMD-13324. Two representative 4x binned tomograms for each acquisition scheme or mouse strain were deposited in EMPIAR under the accession codes: EMPIAR-10771, EMPIAR-10772 and EMPIAR-10773.

The following dataset was generated:

| Author(s) | Year | Dataset title | Dataset URL | Database and Identifier |
|---|---|---|---|---|
| Poege M, Mahamid J, Imanishi SS, Plitzko JM, Palczewski K, Baumeister W | 2021 | Peripheral row of the protein scaffold at rod outer segment disk rims in wild type mice (conventional defocused data). | https://www.ebi.ac.uk/emdb/EMD-13322 | EMDB, EMD-13322 |

*Continued on next page*

*Continued*

| Author(s) | Year | Dataset title | Dataset URL | Database and Identifier |
|---|---|---|---|---|
| Poege M, Mahamid J, Imanishi SS, Plitzko JM, Palczewski K, Baumeister W | 2021 | Central row of the protein scaffold at rod outer segment disk rims in wild type mice (conventional defocused data). | https://www.ebi.ac.uk/emdb/EMD-13321 | EMDB, EMD-13321 |
| Poege M, Mahamid J, Imanishi SS, Plitzko JM, Palczewski K, Baumeister W | 2021 | Central row of the protein scaffold at rod outer segment disk rims in wild type mice (Volta phase plate data). | https://www.ebi.ac.uk/emdb/EMD-13323 | EMDB, EMD-13323 |
| Poege M, Mahamid J, Imanishi SS, Plitzko JM, Palczewski K, Baumeister W | 2021 | Central row of the protein scaffold at rod outer segment disk rims in ABCA4 knockout mice (Volta phase plate data). | https://www.ebi.ac.uk/emdb/EMD-13324 | EMDB, EMD-13324 |
| Poege M, Mahamid J, Imanishi SS, Plitzko JM, Palczewski K, Baumeister W | 2021 | Cryo-electron tomography of rod outer segments in ABCA4 knockout mice acquired with Volta phase plate in focus | https://dx.doi.org/10.6019/EMPIAR-10771 | Electron Microscopy Public Image Archive, 10.6019/EMPIAR-10771 |
| Poege M, Mahamid J, Imanishi SS, Plitzko JM, Palczewski K, Baumeister W | 2021 | Cryo-electron tomography of rod outer segments in wild type mice acquired with Volta phase plate in focus | https://dx.doi.org/10.6019/EMPIAR-10772 | Electron Microscopy Public Image Archive, 10.6019/EMPIAR-10772 |
| Poege M, Mahamid J, Imanishi SS, Plitzko JM, Palczewski K, Baumeister W | 2021 | Cryo-electron tomography of rod outer segments in wild type mice acquired conventionally with defocus | https://dx.doi.org/10.6019/EMPIAR-10773 | Electron Microscopy Public Image Archive, 10.6019/EMPIAR-10773 |

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
