## [Editor Report]

Pöge et al., present a study of the rod outer segment (ROS). These are specialised cilia of rod photoreceptor cells, essential for sensing light cues and initiating the vision process. The authors apply cryo-FIB milling to generate highly preserved rod samples and report high-quality cryo-tomographic data providing new insights into the ultrastructure of the ROS. The work reveals potential molecular scaffolds both in the lumen of the membrane stacks and on the surface of the stack providing the structural basis for ROS crucial ordered ultrastructure. The data presented here will be highly valuable for the field of phototransduction.

---

## [Decision Letter]

**Decision letter after peer review:**

Thank you for submitting your article "Determinants shaping the nanoscale architecture of the mouse rod outer segment" for consideration by *eLife*. Your article has been reviewed by 3 peer reviewers, including Giulia Zanetti as the Reviewing Editor and Reviewer #1, and the evaluation has been overseen by Suzanne Pfeffer as the Senior Editor.

Essential revisions:

1) Please add missing citations to acknowledge previous work and tone down novelty claims (refer to reviewer 2 specific points 1 through 6).

2) The authors assign densities from subvolume averaging to specific complexes based on comparison with data in the literature, but direct experimental validation is not present. All three reviewers note this and it is therefore essential that a) it is made clear the assignments are hypothetical (in text and figures) and b) more support is provided for the model in Figure 5, for example homology modelling and data on PRPH2/ROM1 KO mice.

3) Clarify aspects of the data, for example regarding the curvature of the membrane in the rim complex (see reviewer 1, point 4), and the relationship between various averages of the rim complex (see reviewer 3).

*Reviewer #1 (Recommendations for the authors):*

I am impressed with the data presented in this paper, and I commend the authors on the clarity of the methods section, as well as on the submissions of all maps and raw FSC data.

*Reviewer #3 (Recommendations for the authors):*

The quality of the imaging and subtomogram averaging is of the highest level that can be achieved by cellular cryoET. I have a couple of suggestions to make the figures more clear for the general audience as well as a couple of questions:

For the disk rim, 3 averages are mentioned, namely the whole disk rim, CD focus and peripheral density focus.

1) Need to include in Table 1 the parameters for the whole disk average as well.

2) For clarity Figure 3 should first show the whole disk average currently B and C and then move to the CD focus average currently A. Also the membrane should be annotated in the CD average, for example by segmenting it in different colour. This figure should also include PR focus for completion.

3) Can the author comment on why the TMD is weaker for the PR compared to CD. Not visible at all in Figure 5 A/B left.

4) The authors suggest in their model that the CD and PR are made of the same complex protein, if that is the case why they didn't merge the data for CD and PR to have one average? are the differences they are expecting due to flexibility be observable in the resolution achieved?

As mentioned, a weakness of this manuscript is the direct identification of the densities observed.

1) The authors provided an average from a transgenic mouse without Abca4, however, that average is of lower resolution. Can the authors rule out any differences between WT and the mutant, e.g. in Figure 4 lower panels some differences can be seen. Were the averages in Figure 4 filtered to the same resolution?

2) Can the authors build homology models for PRPH2/ROM1 and GARP2 and try to fit in their map?

Data on PRPH2/ROM1 and GARP2 KO could have strengthen this manuscript substantially.

---

## [Author Response]

Essential revisions:1) Please add missing citations to acknowledge previous work and tone down novelty claims (refer to reviewer 2 specific points 1 through 6).The novelty of the current observation of two types of links is overstated, for example, in the abstract: "Our data reveal the existence of two molecular connectors/spacers which likely contribute to the nanometer scale precise stacking of the ROS disks" (Line 25). In fact, both of these links have been shown before (Usukura and Yamada, 1981; Roof and Heuser, 1982; Corless and Schneider, 1987; Corless et al., 1987; Kajimura et al., 2000). These previous studies deserve to be recognized. Of special note is the paper by Usukura and Yamada whose images of the disc rim connectors are by no means less convincing than shown in the current manuscript. On the other hand, the novelty and impact of the data related to peripherin appears to be understated, particularly in the abstract.

We thank the reviewers for this detailed criticism. We changed the abstract line 27 to:

“Our data confirm the existence of two previously observed molecular connectors …”

cite the recommended references in the introduction (lines 54-55), the results (lines 131-132), and the discussion (lines 282/285). To highlight the previous reports, we rephrased the sentence in lines 132-133:

“In agreement with these previous findings, we observed structures that connect membranes of two adjacent disks …”;

the discussion is rephrased in lines 280-281:

“Similar connectors have been observed previously …” and “… and their statistical analysis confirmed the existence of two distinct connector species.”,

and in lines 291-292:

“Based on previous studies combined with our quantitative analysis, we put forward a hypothesis for the molecular identity of the disk rim connector which agrees in part with recent models”.

2) The authors assign densities from subvolume averaging to specific complexes based on comparison with data in the literature, but direct experimental validation is not present. All three reviewers note this and it is therefore essential that a) it is made clear the assignments are hypothetical (in text and figures) and b) more support is provided for the model in Figure 5, for example homology modelling and data on PRPH2/ROM1 KO mice.

We appreciate this critique from the reviewers that helped improve the presentation of our work. First, we tried to model PRHP2 and ROM1 complexes based on molecular dynamics simulations and homolog structures in PDB without success. Hence, we performed structure prediction of PRPH2 oligomers (dimers and tetramers) with ColabFold, a notebook-based environment based on AlphaFold2. Four of the five predicted dimer models resembled the V-shape of the repeats resolved in the disk rim scaffold. Ridged-body fitting of the best scoring model into the repeat of the central density row appears reasonable at the 18 Å resolution of the average (see Figure 1B).

On the other hand, none of the tetramer models fits into a single repeat of the disk rim scaffold or into two neighboring repeats. Furthermore, the tetramer models are significantly bulkier than the negative stain structure of the putative PRPH2-ROM1 tetramer (Kevany et al., 2013).

This result contradicts the previous interpretation of the low-resolution negative stain structure of PRPH2-ROM1 complexes as tetramers, which also has a similar size and shape as the repeats of the disk rim scaffold resolved in our study. Therefore, the predicted model points to an error in our original interpretation and in previous publications where the putative non-covalently bound subunit of PRPH2 and ROM1 was studied. In these publications, particles of similar properties to those reported in Kevany et al., 2013, lay the foundation for the currently accepted PRPH2-ROM1 tetramer model (reviewed in Goldberg et al., 2016). The accompanying results of native gels and the hydrodynamic data that support the tetramer model may have been biased by the detergents used to purify PRPH2-ROM1 complexes (Triton X-100, Amphipol) or the non-globular shape of the complexes.

Therefore, we have introduced major changes to the manuscript. We added an additional Materials and methods section “Structural prediction of PRPH2 oligomers”. We added a supplementary figure and a video showing the fit of the predicted PRPH2 dimer structure into our subvolume average. We toned down our hypothesis that the repeats of the disk rim scaffold are composed of PRPH2-ROM1 tetramers and refer only to PRPH2-ROM1 complexes in the abstract, results and discussion. We now describe the discrepancy between the predicted PRPH2-dimer model and the negative stain structure of putative PRPH2-ROM1 tetramers and conclude that a high-resolution density map is required to elucidate whether the repeats are composed of dimers or tetramers. Furthermore, we changed the discussion of the disulfide bond forming PRPH2-C150 cysteine residues in light of the predicted model.

Another key point raised in the essential revisions point 2 is to make it clear that our assignment of PRPH2-ROM1 complexes to the repeats of the disk rim scaffold are hypothetical. We agree it is important to highlight that we don’t have direct experimental evidence to document this interpretation. We changed the abstract in lines 28-30:

“We further provide evidence that the extreme radius of curvature at the disk rims is enforced by a continuous supramolecular assembly and provide evidence that it is composed of peripherin-2 (PRPH2) and…”.

and the discussion is revised in lines 358-360:

“Hence, we hypothesize that the repeats resolved in our average are smaller PRPH2-ROM1 complexes which oligomerize to form the disk rim scaffold …”.

We also absolutely agree that data on knockout mice could strengthen our manuscript, and we appreciate this suggestion. Four knockouts are of special interest for us. Our putative candidates for the disk rim and disk interior connectors are (i) GARP2 and (ii) PDE6, respectively; and (iii) PRPH2 and (iv) ROM1 are proposed as potential building blocks of the disk rim scaffold. Unfortunately, PRPH2 and PDE6 knockouts lead to complete retinal degeneration. Hence, ROS of these strains cannot be investigated by cryo-ET. The *Cngb 1* locus encodes three proteins, namely, the β-subunit of the CNGC-channel and the spliced isoforms GARP1 and GARP2. To our knowledge there is no knockout of only GARP2. In addition, GARP2 is predicted to be largely unstructured (Batra-Safferling et al., 2006). Therefore, structural modeling of the disk rim connectors is not possible. For *Cngb 1* or ROM1 knockout mice, ROS can form, as revealed by conventional plastic embedded EM (Zhang et al., 2009, Clarke et al., 2000, respectively). However, ROS are destabilized in both strains and their architecture is highly disorganized (Clarke et al., 2000, Zhang et al., 2009). ROM1 KO mice are indeed commercially available, and the analysis of their ROS by cryo-ET is possible, but not feasible within a reasonable time frame for our revisions. We hope the reviewers will find that the structural models now included in the manuscript help substantiate our hypothesis on the identity of the disk rim scaffold.

3) Clarify aspects of the data, for example regarding the curvature of the membrane in the rim complex (see reviewer 1, point 4), and the relationship between various averages of the rim complex (see reviewer 3).

We appreciate that the reviewers made us aware of this somewhat confusing description. Before we answer the specific points of reviewers #1 and #3 we want to clarify the relationship among various averages. We analyzed three datasets in our work, namely, WT_conv_, WT_VPP_ and Abca4^-/-^_VPP_. For every dataset, an average of the disk rim was calculated with a mask focused on four repeats along the central row in the final alignment. Even though the alignment focused only on a small fraction of the disk rim – its central row – the subvolume averages contain the signal of the whole disk rim. Dependent on what we wanted to show, we either displayed the whole, unmasked average or the average with the alignment mask applied. The unmasked average shows best the organization of the whole disk rim in 3 rows of repeats (Figure 3B, C, Figure 4, Video 7), and it was used to measure the repeat length in all three rows and the offsets of the repeats in the central and peripheral rows with respect to each other (Figure 3—figure supplement 2 C). The average in Figure 3A with the alignment mask applied highlights the connections of the central row to the peripheral rows and the TMDs. Apart from the masking, it is the same average as in Figure 3B and C. The transition from whole, unmasked average to masked average is shown in Video 8. For the WT_conv_ dataset, we also calculated an independent subvolume average of the peripheral row. Here, we used a similar representation as for the central density of the WT_conv_ dataset: In Figure 3—figure supplement 3A, we show slices of the whole, unmasked PR average and in Figure 3—figure supplement 3B, C the masked PR average is shown to highlight TMDs and the connections between adjacent repeats. The transition between whole, unmasked average and masked average of the peripheral row is illustrated in Video 9.

To better explain the relationship among averages in the text, we added a new paragraph starting at line 802:

“Even though the resulting CD average was obtained by focusing the alignment on 4 repeats along the central row of density, the average comprised the signal of the whole disk rim scaffold. We used different representations of this average to highlight certain aspects of the disk rim scaffold. The unmasked CD average shows the organization of the three interconnected rows of density (Figure 3B, C and Video 7), while the alignment mask applied to the CD average reveals the signal of the transmembrane densities (Figure 3A, Video 8).”

Furthermore, we changed the legends of Figure 3, Figure 3—figure supplement 3, Figure 4 and Video 7 through 9, accordingly.

Reviewer #3 (Recommendations for the authors):The quality of the imaging and subtomogram averaging is of the highest level that can be achieved by cellular cryoET. I have a couple of suggestions to make the figures more clear for the general audience as well as a couple of questions:

We thank the reviewer for the positive evaluation of our manuscript and the suggestions, which help make our manuscript more understandable.

For the disk rim, 3 averages are mentioned, namely the whole disk rim, CD focus and peripheral density focus.

We analyzed three datasets in our work, namely, WT_conv_, WT_VPP_ and Abca4^-/-^_VPP_. For every dataset, an average of the disk rim was calculated with a mask focused on four repeats along the central row in the final alignment. Even though the alignment focused only on a small fraction of the disk rim – its central row – the subvolume averages contain the signal of the whole disk rim. Dependent on what we wanted to show, we either displayed the whole, unmasked average or the average with the alignment mask applied. The unmasked average shows best the organization of the whole disk rim in 3 rows of repeats (Figure 3B, C, Figure 4, Video 7), and it was used to measure the repeat length in all three rows and the offsets of the repeats in the central and peripheral rows with respect to each other (Figure 3—figure supplement 2 C). The average in Figure 3A with the alignment mask applied highlights the connections of the central row to the peripheral rows and the TMDs. Apart from the masking, it is the same average as in Figure 3B and C. The transition from whole, unmasked average to masked average is shown in Video 8.

1) Need to include in Table 1 the parameters for the whole disk average as well.2) For clarity Figure 3 should first show the whole disk average currently B and C and then move to the CD focus average currently A. Also the membrane should be annotated in the CD average, for example by segmenting it in different colour. This figure should also include PR focus for completion.

We hope, given the previous explanation, it becomes clear that the unmasked CD average is the same as the CD average with alignment mask applied, to which reviewer #3 referred to as whole average and CD focus, respectively. Therefore, data of all four averages shown in our manuscripts is contained in Table 1. Furthermore, we did not rearrange Figure 3. But as suggested, we marked the signal of the inner and outer membrane leaflet in the front views of the isosurface representations of the averages. Hence, we changed Figure 3A and Figure 3—figure supplement 3B.

3) Can the author comment on why the TMD is weaker for the PR compared to CD. Not visible at all in Figure 5 A/B left.

In Figure 5A, B we show an isosurface representation of the whole, unmasked CD average. At the threshold used, we clearly see the TMD of the central row, but do not in the PR. The grey scale values slice in Figure 3B shows faint indications of the PRs TMD. However, the reviewer is correct that the TMD in the PR average are less well resolved than in the CD and appear less ordered (compare video 8 and 9). We think that the central row is in a more stable conformation as it is fixed in position by the PRs on each side. In turn, the PRs which have the support of the scaffold only on one side by the CD, are more flexible. This is supported by the fact that we have 5x more subvolumes for the PR, but achieved a similar resolution as for the CD (Table 1). Nevertheless, focused alignments on the PRs as shown in Figure 3—figure supplement 3 provides clear densities for their TMDs.

4) The authors suggest in their model that the CD and PR are made of the same complex protein, if that is the case why they didn't merge the data for CD and PR to have one average? are the differences they are expecting due to flexibility be observable in the resolution achieved?

We have indeed tried to merge the densities to improve the structural analysis. However, in line with the comment above, the repeats of CD and PR exhibit different conformations as they are linked differently to their neighbors (Figure 3 and Figure 3—figure supplement 3), and therefore this attempt did not result in better averages.

As mentioned, a weakness of this manuscript is the direct identification of the densities observed.1) The authors provided an average from a transgenic mouse without Abca4, however, that average is of lower resolution. Can the authors rule out any differences between WT and the mutant, e.g. in Figure 4 lower panels some differences can be seen. Were the averages in Figure 4 filtered to the same resolution?

The averages in Figure 4 were not filtered to the same resolution, which we agree is necessary for a fair comparison. We adjusted this Figure by filtering all three CD averages to a resolution of 30 Å. Now the similarity among all three averages is even more striking.

2) Can the authors build homology models for PRPH2/ROM1 and GARP2 and try to fit in their map?Data on PRPH2/ROM1 and GARP2 KO could have strengthen this manuscript substantially.

Homology modeling is an exquisite suggestion. First, we tried to model PRHP2 and ROM1 complexes based on molecular dynamics simulations and homolog structures in PDB without success. Hence, we performed structure prediction of PRPH2 oligomers (dimers and tetramers) with ColabFold, a notebook-based environment based on AlphaFold2. Four of the five predicted dimer models resembled the V-shape of the repeats resolved in the disk rim scaffold. Ridged-body fitting of the best scoring model into the repeat of the central density row appears reasonable at the 18 Å resolution of the average (see Figure 14).

On the other hand, none of the tetramer models fits into a single repeat of the disk rim scaffold or into two neighboring repeats. Furthermore, the tetramer models are significantly bulkier than the negative stain structure of the putative PRPH2-ROM1 tetramer (Kevany et al., 2013).

This result contradicts the previous interpretation of the low-resolution negative stain structure of PRPH2-ROM1 complexes as tetramers, which also has a similar size and shape as the repeats of the disk rim scaffold resolved in our study. Therefore, the predicted model points to an error in our original interpretation and in previous publications where the putative non-covalently bound subunit of PRPH2 and ROM1 was studied. In these publications, particles of similar properties to those reported in Kevany et al., 2013, lay the foundation for the currently accepted PRPH2-ROM1 tetramer model (reviewed in Goldberg et al., 2016). The accompanying results of native gels and the hydrodynamic data that support the tetramer model may have been biased by the detergents used to purify PRPH2-ROM1 complexes (Triton X-100, Amphipol) or the non-globular shape of the complexes.

Therefore, we have introduced major changes to the manuscript. We added an additional Materials and methods section “Structural prediction of PRPH2 oligomers”. We added a supplementary figure and a video showing the fit of the predicted PRPH2 dimer structure into our subvolume average. We toned down our hypothesis that the repeats of the disk rim scaffold are composed of PRPH2-ROM1 tetramers and refer only to PRPH2-ROM1 complexes in the abstract, results and discussion. We now describe the discrepancy between the predicted PRPH2-dimer model and the negative stain structure of putative PRPH2-ROM1 tetramers and conclude that a high-resolution density map is required to elucidate whether the repeats are composed of dimers or tetramers. Furthermore, we changed the discussion of the disulfide bond forming PRPH2-C150 cysteine residues in light of the predicted model.

We also absolutely agree that data on knockout mice could strengthen our manuscript, and we appreciate this suggestion. Four knockouts are of special interest for us. Our putative candidates for the disk rim and disk interior connectors are (i) GARP2 and (ii) PDE6, respectively; and (iii) PRPH2 and (iv) ROM1 are proposed as potential building blocks of the disk rim scaffold. Unfortunately, PRPH2 and PDE6 knockouts lead to complete retinal degeneration. Hence, ROS of these strains cannot be investigated by cryo-ET. The *Cngb 1* locus encodes three proteins, namely, the β-subunit of the CNGC-channel and the spliced isoforms GARP1 and GARP2. To our knowledge there is no knockout of only GARP2. In addition, GARP2 is predicted to be largely unstructured (Batra-Safferling et al., 2006). Therefore, structural modeling of the disk rim connectors is not possible. For *Cngb 1* or ROM1 knockout mice, ROS can form, as revealed by conventional plastic embedded EM (Zhang et al., 2009, Clarke et al., 2000, respectively). However, ROS are destabilized in both strains and their architecture is highly disorganized (Clarke et al., 2000, Zhang et al., 2009). ROM1 KO mice are indeed commercially available, and the analysis of their ROS by cryo-ET is possible, but not feasible within a reasonable time frame for our revisions. We hope the reviewers will find that the structural models now included in the manuscript help substantiate our hypothesis on the identity of the disk rim scaffold.